# Automated detection of complex zebrafish seizure behavior at scale
Paige Whyte-Fagundes [1,5] ✉, John Efromson [2,5], Anjelica Vance[1,3], Samuel Carpenter[2,4], Aurélien Bègue[2], Aloe Carroll [1], Thomas Jedidiah Jenks Doman[2], Mark Harfouche[2] & Scott C. Baraban [1,3] ✉

Convulsive seizure behaviors are a hallmark feature of epilepsy, but automated detection of these events in freely moving animals is difficult. Here, we employed a high-resolution multi-camera array microscope with high-speed video acquisition and custom supervised machine learning (ML) for automated detection of larval zebrafish between 3- and 7-days post-fertilization (dpf). We assessed data from over 2700 zebrafish either exposed to a chemoconvulsant (pentylenetetrazole, PTZ) or genetic zebrafish lines representing Developmental Epileptic Encephalopathy (DEE) syndromes. Using eight-point skeletal body pose estimation for tracking individual larvae arrayed in a 96-well format, we report reliable, quantitative and age-dependent changes in maximum swim speed, as well as eye-, head- and tail- angle kinematics. Finally, we employed an ML-based algorithm to automatically identify normal and abnormal behaviors in an unbiased manner. Our results offer a robust framework for automated detection of zebrafish seizure-associated behaviors.

Seizures manifest as stereotypical behaviors including muscle spasms, sustained rhythmic jerking, focal stiffening and behavioral arrest[1]. Accurate identification of these movements is critical for understanding seizure generation and developing new therapeutics[2–5]. This understanding necessitates use of experimental animal models of seizures and epilepsy[6–8]. These range from spontaneous models of epilepsy (dogs, baboons and domoic-acid poisoned California sea lions)[9,10] to acquired and acute experimental models (cats, rabbits, monkeys and rodents)[7,11–13]. Of these varied species options, rodents emerged as the most widely used animal model in the epilepsy field[14–16]. A long-established standard for evaluation of rodent seizure behavior is a semi-quantitative observational scale developed by Racine in 1972 using an amygdala-kindling model[17]. In rodents, seizure behaviors include whisker twitching, stiffened tail, rearing and forelimb clonus. However, unbiased analysis of such behaviors can be challenging, especially as seizures in unrestrained animals consist of elaborate and variable movements organized at multiple timescales. Automating this observation-based system and scaling to automatically monitor hundreds of experimental animals simultaneously also presents a daunting challenge. Although recent technical advances are beginning to address these issues in rodents, including adaptation of deep-learning-based platforms (such as MoSeq)[2] that incorporate 3-dimensional cameras and unsupervised machine learning[4], scalability with rodents remains limited.

Seizure behavior and antiseizure medication (ASM) discoveries historically employ animal models using chemical convulsant agents[15] such as PTZ, a $GABA_A$ receptor antagonist, first described in the 1950s[18,19]. This model is simple and reliably elicits acute seizures in mice, cats, rabbits, gerbils, monkeys and rats[20]. PTZ is water-soluble, facilitating adaptation of this model to larval zebrafish (*Danio rerio*)[21–32] where drugs dissolved in bathing medium are rapidly absorbed directly through the skin[33]. As a small vertebrate, zebrafish offer significant advantages of scale over other experimental animal seizure models[34] and are amenable to large-scale drug screening[35–38]. Their translational value is further enhanced by multifaceted parallels to humans at genetic, cellular, and behavioral levels. Zebrafish share a conserved neuroarchitecture, including forebrain (telencephalon), midbrain (mesencephalon) and hindbrain (rhombencephalon) regions that are functionally analogous to those in mammals[39–43]. From a functional perspective, specific cell types necessary for generation of network activity are present from 2 days post-fertilization (dpf) and include excitatory glutamatergic[44], inhibitory GABAergic[41,44], monoaminergic (dopamine, noradrenaline, serotonin, histamine) neurons[41,45] and non-neuronal (astrocytes, microglia) cells[46,47]. Larval zebrafish can exhibit electrographic seizure activity with similarities to human ictal events, including high-amplitude, synchronized spike discharges, and seizure-like rhythmic local field potential (LFP) patterns[21,22,48]. Their amenability to genetic manipulation enables real-time functional validation of epilepsy-associated genes

[1]Department of Neurological Surgery & Weill Institute for Neuroscience, University of California, San Francisco, CA, USA. [2]Ramona Optics Inc., Durham, NC, USA. [3]Helen Wills Institute of Neuroscience, University of California, Berkeley, CA, USA. [4]Department of Computer Science, Duke University, Durham, NC, USA. [5]These authors contributed equally: Paige Whyte-Fagundes, John Efromson. ✉e-mail: paige.whytefagundes@ucsf.edu; scott.baraban@ucsf.edu

identified in patients[49,50], thus facilitating genotype-phenotype studies in a living vertebrate system. Further, zebrafish-based drug discovery led to the identification of candidate therapies, such as clemizole, lorcaserin and fenfluramine[37,38,51,52], which progressed to clinical trials for DEE patients. Recent advances in high-resolution imaging and machine learning have further enhanced the utility of this model. For example, studies of normal larval swim behavior[53,54] highlight the potential to quantify complex motor and non-motor manifestations relevant to seizure phenotyping.

Analysis of behavior is a critical functional readout of brain activity. Here we describe a robust, unbiased and high-throughput approach to automatically monitor complex behavior in larval zebrafish. PTZ, a traditional chemoconvulsant, was used to elicit seizures in zebrafish during early development (3- to 7- dpf). Using a multi-camera array microscope (MCAM) that tethers 24 high-resolution cameras to capture an entire 96-well plate at 312 megapixel (MP) resolution[55,56], we acquired imaging data on freely moving larval zebrafish at 160 frames per second (fps). With sufficient spatial resolution to discern individual larval posture, this approach enabled precise measurement of head, eye and tail location. Abnormal high-speed convulsive behaviors were identified at maximum swim speeds of 120 mm/sec in a single well. Significant age-dependent, seizure-associated changes in eye, head and tail angle kinematics (not fully appreciated using single-camera low-resolution systems) are described. Additionally, we developed an automated ML-based behavioral classification algorithm to detect and assess the diverse and dynamic seizure-like activities of larval zebrafish at scale. Although the complexity and variability of seizure-like behaviors in zebrafish larvae pose unique challenges for behavioral analysis, we successfully generalized our model to genetic zebrafish models representing DEEs. Total activity analyses revealed increased swim movement in both *pnpo* and *scn1lab* mutants, and our algorithm reliably identified and distinguished seizure-like behavioral motifs (including rapid darting, whirlpool, and clonus-like tail beats specific to *scn1lab* larvae). Adapting these technologies to analyze clinically relevant seizure behaviors could significantly improve ASM evaluation by enabling more accurate, scalable behavioral screening and accelerate drug discovery.

## Results

### High-resolution high-speed video acquisition facilitates automated detection of quantifiable seizure behaviors

Larval zebrafish seizures were initially described as a simple repertoire of swim movements culminating in full-body convulsions[22]. These behaviors were tracked using a single low-resolution camera and a single-point detection system at an acquisition speed of 25 fps (Supplementary Fig. 1A, left). From these observations, a semi-quantitative scale capturing three stages of behavior extending from a modest increase in swim behavior (stage 1), to rapid whirlpool-like movements along the well perimeter (stage 2) to convulsive movements followed by a brief loss of posture (stage 3) was established[22]. However, this qualitative scale naturally incorporates some investigator-bias and previous first-generation video acquisition tools do not offer adequate resolution of high-speed and fine motor movements across the entire 96-well plate. Here, we used a multi-camera array microscope with 24 high-resolution cameras to record freely swimming larvae at 160 fps (Supplementary Fig. 1a, right) coupled with pose estimation of eight key-points per larva (Supplementary Fig. 1b; Supplementary Video 1). Accuracy evaluation of the pose estimation model suggested inferred coordinate errors ranging from 55 to 130 microns depending on the key-point coordinate (Supplementary Fig. 2). First, we established a detectable movement rate of 120 mm/sec as a cutoff for swim speeds reached by an individual larva in a single 0.32 cm² well by first plotting speeds of 1632 larvae recorded on a log scale across 48,000 frames (Supplementary Fig. 1c). Second, to determine tracking noise for *x* and *y* components of all estimated key-points, mean sigma was plotted for each (Supplementary Fig. 1d). Third, to achieve higher levels of tracking data output accuracy these sigma values were used as an input in a wavelet denoising algorithm, along with center-of-mass (COM) and speed threshold filtering (Supplementary Fig. 1e). Representative tracking plots show where individual larvae travel

within a well and encode speed thresholds via color. Larvae moving between 0 and 50 mm/s are presented along a gradient of blue-to-red with larvae moving faster than 50 mm/s shown in green (Supplementary Fig. 1e).

As zebrafish larvae are rapidly maturing in the first week post-fertilization[57,58], we first identified the range of behavioral seizure activity possible at 3, 5 and 7 dpf (Fig. 1a) e.g., commonly used developmental stages in most zebrafish neuroscience studies. At baseline, spontaneous swimming is sporadic and larvae primarily swim at speeds lower than 50 mm/s (Fig. 1b, c, *baseline*)[59,60]. Although infrequent, brief high-speed darting movements are possible. At the first time point, 5 min after exposure to 15 mM PTZ, zebrafish larval swim movement increases dramatically to cover the entire well arena at speeds much greater than 50 mm/s (Fig. 1b, c, *TP1*). To quantify zebrafish movement within each well, we calculated an activity score (summation of movement over recording duration based on change in pixels from frame-to-frame presented on a log scale) (Supplementary Fig. 3). Plotting instantaneous activity across time for a single representative larva during each recording epoch highlights how the peak activity metric increases nearly 3-fold from baseline with PTZ (Fig. 1c). These plots also show that activity during the first PTZ timepoint1 (TP1) is characterized by a "bursting" pattern not seen at baseline that becomes more frequent with continued PTZ exposure (TP2). Zebrafish larvae at all three ages exhibit a significant increase in total activity from baseline with PTZ (Wilcoxon test: $p < 1E^-15$ for each population at TP1 and TP2) (Fig. 2a). Likely owing to increased episodes of seizure related posture loss with continuous PTZ exposure, larvae at 5 or 7 dpf demonstrate a slight decrease in total activity from TP1 to TP2 (5 dpf, $p < 1E^-15$; 7dpf, $p = 7.9E^-5$). Consistent with total activity metric scores increasing with PTZ, larvae also display significant increases in total distance traveled (m) at 3, 5 and 7 dpf (Wilcoxon test: $p < 1E^-15$ for each population at TP1 and TP2) (Fig. 2b).

Single-camera, single-point detection systems using an algorithm for locating objects darker than background most commonly report total distance traveled as the primary metric for identifying seizure-associated behavior[61–63]. Because convulsive seizures are invariably associated with high-speed swim movements, we previously described a custom MATLAB script for single-point detection systems to detect seizure-associated swim activity at speeds greater than 28 mm/s[49]. However, given the 25-fps limitation and low-resolution camera used on these systems, this threshold may underestimate swim velocities reached during a convulsive seizure in a 0.32 cm² well. With video acquisition at 160 fps and camera resolution up to 312 MP we here observed baseline larval swim events that can exceed 50 mm/s (Fig. 3a). However, these speeds are rare at baseline and reached only in brief episodes. In stark contrast, PTZ was invariably associated with sustained high-speed convulsive-associated behavior that consistently reach speeds of 110–120 mm/s in freely swimming larvae at 3, 5 and 7 dpf at both TP1 and TP2 (Fig. 3a). Probability density plots show maximum speeds reached for all 3, 5 and 7 dpf larvae, confirming that PTZ exposure increases maximum swim speeds from baseline to TP1 and TP2. These plots also reveal a slight decrease in maximum swim speeds during TP2 compared to TP1, suggestive of less whirlpooling (stage 2 behaviors) and more convulsive activities (stage 3 behaviors) (Fig. 3b).

### Advanced swim kinematic measurements reveal a wider range of seizure behaviors

Understanding zebrafish swim kinematics contributes to our overall understanding of how the brain controls movement[59,64]. Because epileptic seizures are defined by a repertoire of complex behaviors[1,4], uncovering novel swim kinematics in seizing larval zebrafish will contribute to our understanding of epilepsy. Here we used pose estimation algorithms with 8-points aligned along the zebrafish larvae skeleton, for unbiased quantification of freely swimming larval behavior during PTZ exposure. Focusing on the interaction of select key-points, we first analyzed how seizures are associated with significant changes in tail and heading angles. Points located on the snout and center of the body were used to create heading and body axis vectors, and the center and caudal fin points create a tail vector (Fig. 4a). The tail vector was used to calculate changes in tail angles with respect to

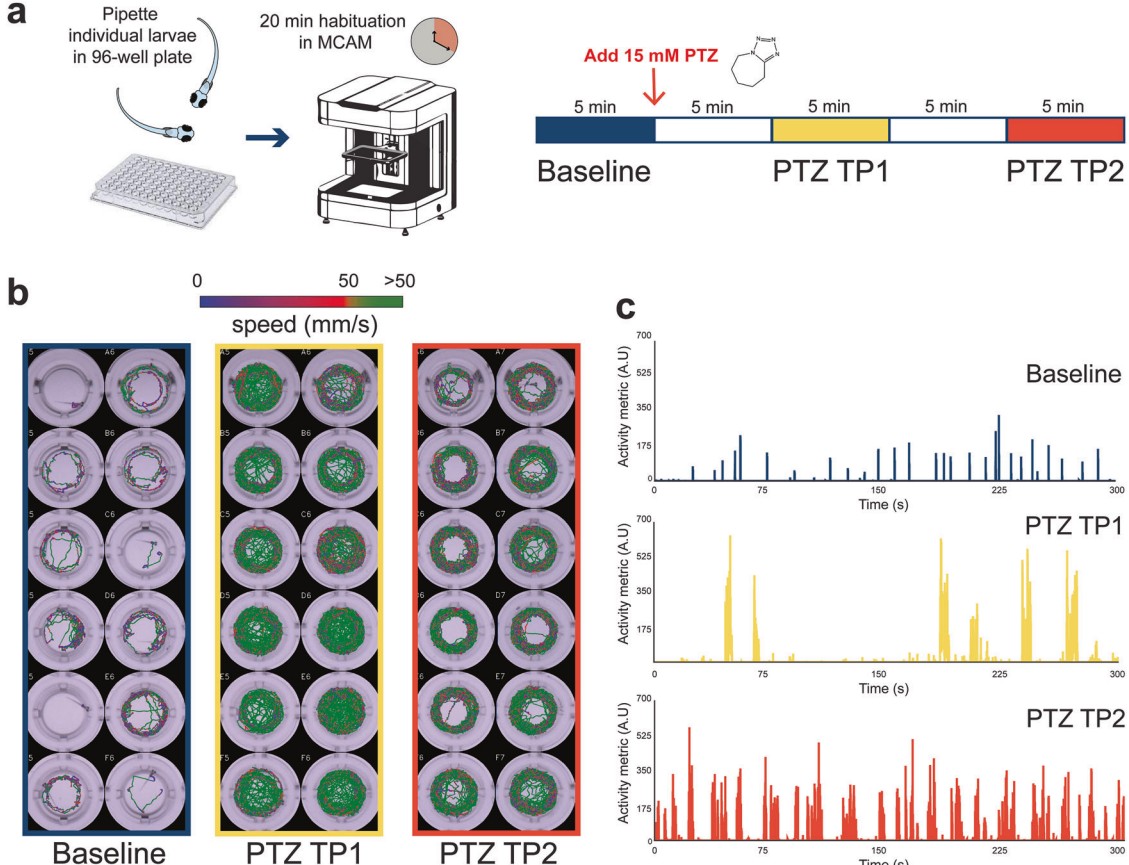

**Fig. 1 | PTZ seizures. a** Schematic for MCAM acquisition protocol; videos acquired at 160 fps in 5 min epochs. 20 min continuous PTZ exposure with two video acquisitions at timepoint 1 (TP1) and timepoint 2 (TP2). **b** Sample tracking plots across 12 wells of representative larvae at Baseline (blue), PTZ TP1 (yellow) and PTZ TP2 (orange). Tracking plots are color coded to indicate speed from blue (slow) to red (faster) to green (>50 mm/s). Supplementary Fig. 3 shows activity metric values corresponding to the wells shown (**b**). **c** Representative activity metrics plotted over a 5 min recording for a single larva at baseline, PTZ TP1 and PTZ TP2.

body axis vector as zebrafish larvae display a wide range of tail angles from 0°, when larvae are in a normal fully extended position, to 180°, when larvae are curled, and the tail tip touches the snout (Fig. 4b). Heading angle changes were calculated by measuring angle difference of the heading vector on a frame-to-frame basis according to position of an individual larva in the well (Fig. 4c). Quantitative assessment of head (Fig. 4d) and tail (Fig. 4e) angle changes in larvae at 3, 5 and 7 dpf revealed greater alterations in these kinematic measures during PTZ compared to baseline. The most pronounced changes were observed in larvae at later developmental stages.

In late stages of seizure-associated behavioral progression (e.g., stage 3), larvae exhibit sinusoidal full-body convulsions followed by periods of posture loss[22]. To automatically detect these movements, we used two keypoints positioned on the eyes to calculate inter-eye distance, serving as a proxy for posture loss (Fig. 5a, b). When larvae are stationary or swimming upright, both eyes are visible, with an inter-eye distance of approximately 200 μm. This distance decreases to approximately 0 μm when only one eye is visible i.e., a larva with post-seizure posture loss. At 3 dpf, posture loss as measured by inter-eye distance is modestly decreased with PTZ from baseline (Fig. 5c) (Wilcoxon test: $p < 0.003$). However, these PTZ induced reductions in inter-eye distance become more significant with development at 5 and 7 dpf (Fig. 5c) (Wilcoxon test: $p < 1E^-15$ for each population at TP1 and TP2). Representative plots of inter-eye distance over time for the entire 5-min recording is shown for baseline and PTZ timepoint 2 in Fig. 6a. Note the small fluctuations in inter-eye position at baseline when larva is largely stationary or in a normal burst and glide swim mode (Fig. 6a, top trace). In contrast, during full-body convulsions in PTZ we observed abrupt changes in inter-eye distance as larvae exhibit erratic twists and turns within the well (Fig. 6a, bottom trace). Here, we performed parallel experiments combining

local field potential recording (LFP) of electrical activity with fast fiber photometry to monitor brain $Ca^{2+}$ dynamics using a genetically encoded calcium indicator under a pan-neuronal promoter (*Tg(HUC:H2B-GCaMP6s)*). This approach enabled us to capture hypersynchronous neural activity and highlight the temporal similarities with the motor manifestations of this activity exhibited during changes in inter-eye distances (Fig. 6b). As the optic fiber is placed outside the larvae and reports $Ca^{2+}$ dynamics only for GCaMP-expressing neurons as a proxy for neural activity, it provides a sensitive non-invasive measure of brain-specific activity free of movement artifact. To capture $Ca^{2+}$ signals, larvae were positioned dorsal side up in agarose and the optic fiber (100 μm core, 0.22 NA) was placed above forebrain near anterior optic tectum. To simultaneously monitor electrical activity, a microelectrode was inserted into optic tectum. Like baseline inter-eye distance plots in panel 6b (top), baseline LFP + photometry data showed little to no electrical or $Ca^{2+}$ activity. However, resembling the inter-eye PTZ trace in panel 6b, we observed large abrupt $Ca^{2+}$ signal fluctuations synchronized with voltage deflections in LFP typical of ictal-like seizure events followed by a post-ictal electrical signal depression (Fig. 6c, *LFP trace*); $Ca^{2+}$ sensors decay more slowly but accurately capture these seizure event onsets (Fig. 6c, *Delta F/F trace*).

### Evaluation of a common antiseizure medication, valproate

Seizing zebrafish larvae facilitate high-throughput drug screening at a scale not possible using rodent models[65–68]. Here, we determined whether measurements described above could be used to identify an ASM (valproate, VPA), widely used in clinical practice[69,70]. First, we used our functional LFP plus fiber photometry assay to confirm VPA acts to suppress PTZ seizure activity at the level of the central nervous system (CNS). As expected[21–23],

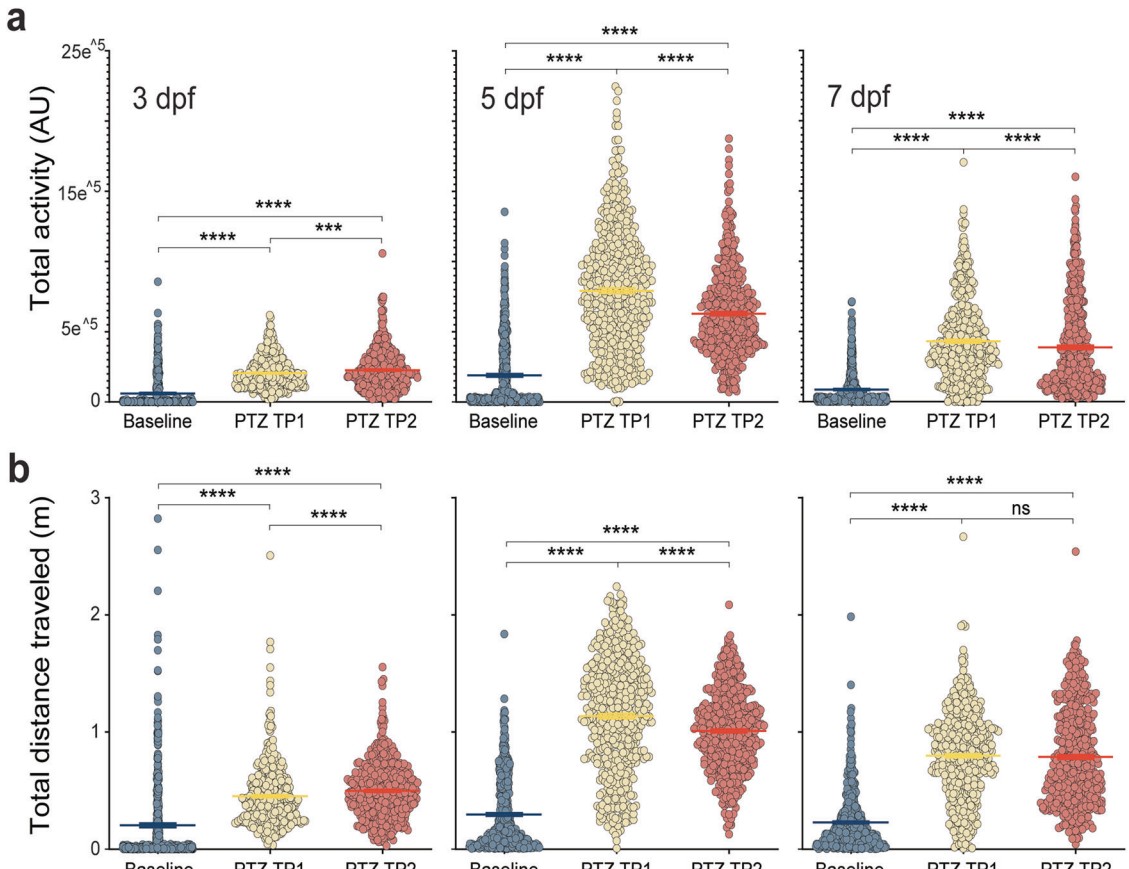

**Fig. 2 | Increased swim movement with PTZ. a** Total activity (AU) shown in log10 scale for each larva recorded at baseline (blue), PTZ TP1 (yellow) and PTZ TP2 (orange) at 3 (left, $N = 473$), 5 (center, $N = 547$) and 7 (right, $N = 546$). Mean shown as straight line. Wilcoxon signed rank ***$p < 0.0005$, ****$p < 0.0001$. **b** Total distance traveled (m) during each recorded epoch for each larva. Mean shown as straight line.

PTZ-induced seizure-like electrographic and $Ca^{2+}$ events were suppressed following VPA exposure in agarose-immobilized larvae (Fig. 7a). Next, to determine whether video acquisition using MCAM, could be used to identify an ASM, we tested larvae randomly plated in a 96-well array. As anticipated, VPA decreased swim movement (Fig. 7b), and significantly reduced average total activity (Fig. 7c, left) and average total distance traveled (Fig. 7c, right) (Wilcoxon test: $p < 1E^{-}15$ for each metric). These results validate a strategy for future drug screening using the MCAM system.

## ML-driven ethograms for unbiased detection of seizure behaviors

Machine learning driven analysis of rodent behaviors are beginning to discover previously hidden phenotypes in freely behaving animals[2,71]. Gschwind et al.[4] recently adapted an ML-assisted analysis to mouse models of epilepsy. However, this approach was limited to a single seizing rodent and does not scale easily. Here, we developed ML algorithms (see Methods) tailored to individual zebrafish behavior in a 96-well plate format. Briefly, video frames were segmented to 60-frame windows, equivalent to 0.375 s of acquisition (Fig. 8a). The first frame of each window underwent egocentric alignment through a series of transformations and translations, which were then applied to the successive 59 frames. Over 400 extracted video clips were labeled for five different behavioral classifications: (i) stationary (no movement), normal swim (brief burst and glide forward movement), whirlpool (rapid swimming along the well edge), convulsion (fast sinusoidal whole-body movements with large right-left tail bends or corkscrew vertical swimming) and posture loss (larvae on side); see Supplementary Videos 2–6. These behaviors were used to train a random forest classification (RFC) model (Fig. 8b). We also evaluated k-nearest neighbors and support vector

classifier algorithms (Supplementary Fig. 4a–c); however, the optimized RFC model best classified these behaviors, achieving a F1 score of 0.872 and an average classification accuracy of 0.874 (Supplementary Fig. 4d, e). Next, we used this model to automatically classify behaviors occurring in each recording epoch across the entire 5 dpf 576 larvae dataset. Results were first visualized as pie charts for each 96-well plate, with one chart per well representing the percentage of each classified behavior during the recording period for each larva (Fig. 8c). Our ML-based analysis of high-resolution video acquisitions from 5 min recording epochs during PTZ exposure revealed robust increases in traditional seizure-like behaviors previously scored as Stage 2 or 3 activity (e.g., whirlpool, convulsion and posture loss)[22]. Swim activity seen during baseline recordings in embryo media primarily consist of 'stationary' and 'normal' burst and glide behaviors. Next, ethograms were used to plot the occurrence of each of these five behaviors over time, aligned with plots for scalar metrics of displacement (m), inter-eye distance (mm), heading and tail angle changes (deg) over time (Fig. 8d). Representative ethograms for baseline, TP1 and TP2 highlight the increased occurrence of whirlpooling, convulsions and posture loss with PTZ. Alongside observed differences between baseline and PTZ with respect to frequency of seizure-like behaviors over time, ML automatically identified seizure episodes which are characterized by a sequential progression from stationary (dark blue) to normal swim (light blue), to whirlpool swimming (orange), to convulsive behavior (red) culminating in posture loss (purple). Finally, representative 75 s segments for the seizure-like behaviors at TP2 further illustrate how ML identified metrics in the ethogram correspond to scalar metrics (Fig. 8e). Each of these behavioral metrics depicts a temporal pattern of activity resembling that seen with monitoring of brain activity patterns (compare Figs. 6c and 8e).

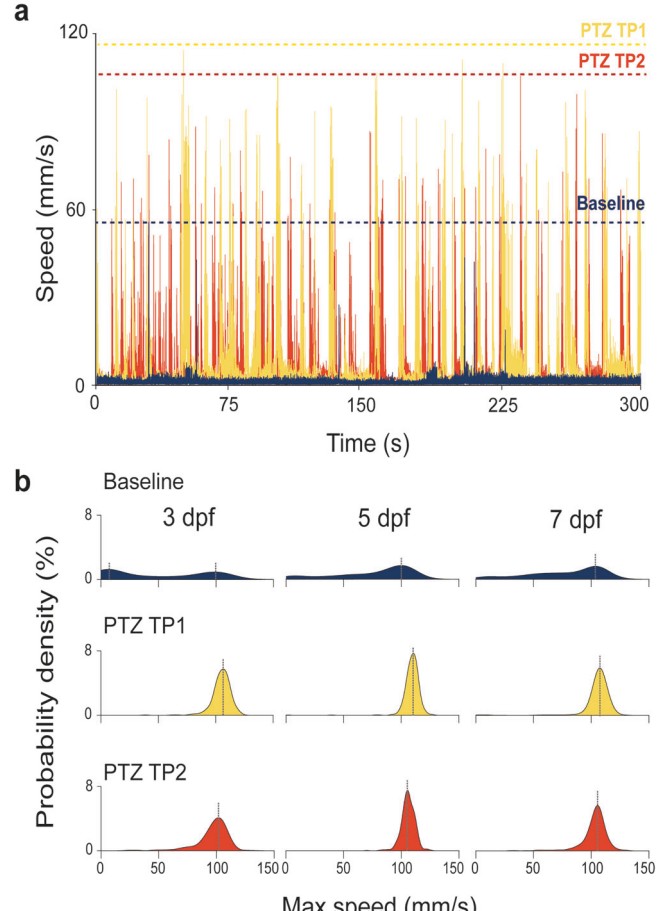

**Fig. 3 | Increased swim speed with PTZ. a** Speed plotted over time for one representative larva at 5 dpf showing the difference in speeds during each recording epoch with dotted lines indicating maximum speeds reached during each epoch.
**b** Probability density plots for maximum speeds (mm/s) reached by 3, 5 and 7 dpf larvae showing an increase in max speeds during TP1 and TP2 from baseline.

## Automated detection of seizure behavior in genetic epilepsy zebrafish models

Next, we examined spontaneous swim behavior using MCAM with six different zebrafish models representing a spectrum of DEEs associated with single-gene mutations. Stable zebrafish lines representing *GABR3*, *PCDH19*, *PNPO*, *SCN1A*, *SCN8A* and *SYNGAP1B* were generated using CRISPR-Cas9 editing[37,49] and raised to F10 generation or later. LFP recordings have reported spontaneous ictal-like electrographic seizure events in *scn1lab* and *pnpo* zebrafish mutants[49,72–74]. Adult heterozygote breeders were crossed to generate clutches of wild type, heterozygote and homozygote mutants which were then randomly selected at 5 dpf and placed individually in a 96-well plate. To quantify zebrafish movement within each well, we calculated an activity score and all larvae were genotyped *post hoc*. A significant increase in total activity was noted for homozygote *pnpo*[−/−] and *scn1lab*[−/−] larvae compared to WT sibling controls (Fig. 9a) (unpaired t-test: $p < 1E^{-}15$ for each population). Next, we used our trained RFC model for automatic behavior classification of *scn1lab* mutants. A representative ethogram highlights increased occurrence of behavior classified as whirlpooling, convulsion and posture loss (Fig. 9b); segments for these seizure-like behaviors illustrate how ML identified metrics in the ethogram correspond to scalar metrics. A sequential progression from stationary (dark blue) to normal swim (light blue), to whirlpool swimming (orange), to convulsive behavior (red) culminating in posture loss (purple) can be seen in some portions of the *scn1lab* ethogram, like that observed for acute PTZ-induced seizures (compare Figs. 8d and 9b).

## Discussion

Deciphering complex behaviors offers a window into our understanding of the brain. Recent advancements in adapting machine learning to analysis of behavior in experimental animal models is beginning to offer new insights into social behavior[75], cerebellar contributions to coordinated locomotion[76], sex-specific behavior[77] and naturalistic self-motivated behavior[78]. Applied here, for the first time, to a widely used acute seizure zebrafish model[21–23,25,29,79–82], as well as genetically modified epilepsy models, we provide a framework to further our understanding of epilepsy (e.g., a neurological disorder marked by recurrent spontaneous seizure behaviors) and aid in future high-throughput drug screening efforts. Our studies revealed sensitive and unique kinematic measurements (head and tail angle changes, inter-eye distance and overall activity) derived from a combination of high-resolution imaging, fast video acquisition, and unbiased ML algorithms. Together these measurements enabled automated detection of complex seizure-like behaviors in larval zebrafish at scale.

Behavioral manifestations of seizures are a hallmark feature of epilepsy patients as well as experimental animal models. Largely described in a vast literature on this topic based upon human observation, we are now beginning to see the first applications of unbiased ML-based approaches to automatically detect and define these complex CNS-generated behaviors[4]. This approach utilizes pose estimation as a computational method for measuring geometrical body configurations. This methodology is enhanced by deep neural networks that allow for more precise, marker-less tracking of skeletal or contour models in experimental animals[83,84]. While existing pose estimation tools like DeepPoseKit[85], ZebraZoom[86] and DeepLabCut[87] are effective for naturalistic behavior tracking, application to neurological disorders, particularly for seizure detection in larval zebrafish only several millimeters in length, is lacking. Here, we developed ML-based methods for monitoring complex larval behaviors, with a particular example shown here for epileptic seizures. High-speed acquisition and image resolution afforded by the multi-array camera microscope[55] facilitated robust pose estimation in freely swimming larval zebrafish arrayed individually in a 96-well plate that would not be possible on a single-camera low-resolution acquisition system or fluorescent plate reader[88]. Single-camera systems[21,24,51,72,82] utilize simple measurements like total distance moved to capture gross larval seizure behaviors. This approach was confirmed and extended here. Interestingly, capturing video data at 160 fps revealed convulsive seizure behaviors reaching much faster swim speeds than previously recognized at 25 fps acquisition[24,49]. The high-resolution image acquisition possible also allowed for precise detection of the large (up to 180°) tail curling episodes associated with these convulsive-like behaviors as well as the rapid changes in head angle. Further, selecting key-points along a larval skeleton model enabled advanced kinematic analysis, including accurate measurement of tail and heading angles, and distance between the eyes over time. These refined and higher order kinematic measures enabled automated and reproducible seizure detection in several thousand PTZ-exposed larvae between 3 and 7 dpf. It is interesting to note that ML detection of behavioral seizure-like patterns over time closely resemble temporal traces of abnormal brain activity using a combination of electrophysiology and $Ca^{2+}$ imaging (see Figs. 6 and 7).

Complex PTZ evoked seizure behaviors are severe and follow a stereotypical sequence from stationary to normal swim, to whirlpool swimming, to convulsive behavior culminating in posture loss that may represent a larval form of tonic-clonic convulsion[22,29]. This sequence is present in the ML ethograms and resembles how this acute seizure model progresses through Racine-like behavioral stages[17,22]. Application of this ML approach also revealed complex behavioral movement patterns in a genetic model of Dravet syndrome (*scn1lab* mutants) that were not fully appreciated in publications using single-camera systems[37,51,89,90]. In *scn1lab* mutants, spontaneous seizures characterized by prominent whirlpool, brief convulsion or loss of posture events in the ethogram occurred in a more random pattern than that observed with PTZ. Interestingly, this pattern was not observed in controls (WT AB or WT siblings) and may recapitulate the spectrum of seizure types (atonic seizures, brief myoclonic seizures and

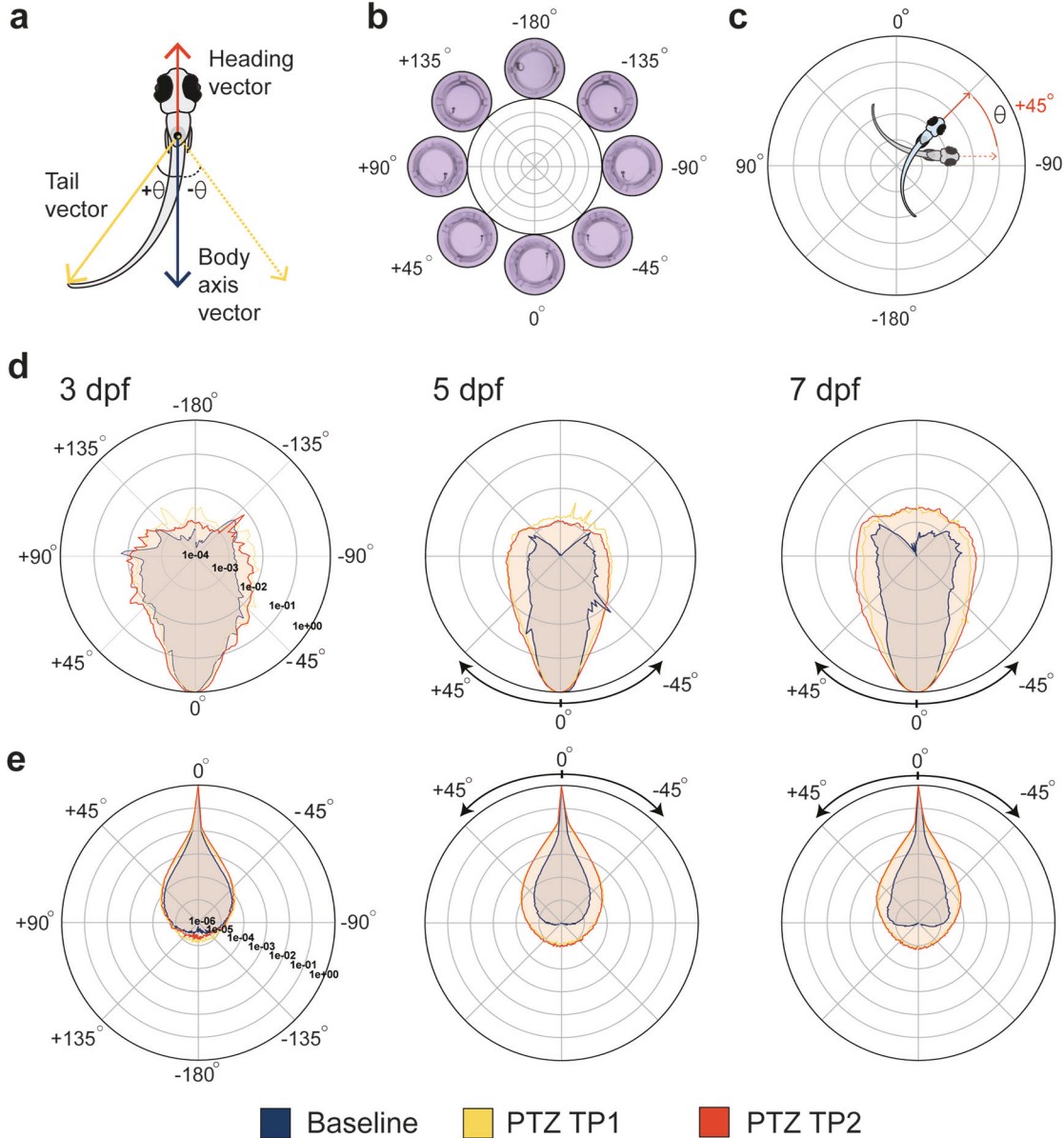

**Fig. 4 | Tail and head angle with PTZ. a** Larval schematic showing heading (orange), tail (yellow) and body axis (blue) vectors used to calculate change in tail angle indicated by Θ, measured between tail and body axis vectors. **b** Representative images of zebrafish larvae displaying examples of different calculated tail angles. **c** Schematic indicating how heading angle change is calculated. Larva in gray in the previous frame is compared to larva in blue in the current frame and heading angle difference is computed. **d** Distribution of tail angle measurements displayed at each recording epoch for 3 (left, N = 50.4 million), 5 (center, N = 71 million) and 7 (right, N = 71 million) dpf larvae showing PTZ induces greater tail angles at all ages. **e** Distribution of heading angle changes with PTZ at all ages.

tonic-clonic convulsions) seen in this patient population[91–94]. Furthermore, unbiased analysis of total activity across six different zebrafish DEE models proved to be a sensitive measure of epilepsy phenotypes as two genetic models (*pnpo* and *scn1lab*) previously shown to exhibit electrographic seizure activity were successfully distinguished, while models with milder or no reported electrographic activity showed no significant differences from controls. Further, we found no evidence of a previously reported hyperexcitability phenotype in *pcdh19* mutants[95]. These data suggest that higher-resolution MCAM imaging capabilities coupled to novel metrics and machine learning algorithms may offer sensitive and deeper computational phenotyping of seizure-like behavior in zebrafish genetic models of epilepsy.

The acute PTZ model in zebrafish larvae serves as a reliable platform for seizure analysis, consistently inducing prominent convulsive behaviors. This made it ideal for training our innovative ML classifier, and enabled development of higher-order kinematic metrics that advance automated detection of the complex behavioral repertoire associated with seizures.

Importantly, by coupling high-resolution, fast-acquisition imaging with ML, we established a scalable platform that is sensitive enough to detect both chemically-induced and spontaneous seizure phenotypes. However, the high-resolution acquisitions required to perform these analyses on a 96-well plate are limited by computing power currently available. For example, current hardware presents physical limitations related to the speed at which central processing unit (CPU) process instructions, the amount of memory available and the size of storage devices which together resulted in the relatively brief 5-minute acquisition epochs used here. Since zebrafish move in three dimensions and images capture only two, the current system may not fully represent the well space occupied by a seizing larva. None the less, these tools establish a new standard for behavioral seizure studies in zebrafish. Leveraging the strengths of zebrafish models for high-throughput chemical screens[38,96–99] and rapid functional evaluation of human genomics[49,100] our platform offers a scalable and powerful approach for advancing large-scale zebrafish-based translational neuroscience research.

**Fig. 5 | Inter-eye distance with PTZ. a** Larval head schematic denoting eye points used to calculate inter-eye distances, *d*. **b** Representative frames from recorded larvae showing examples of stationary and posture loss positions. **c** Plots of average inter-eye distance (mm) for each larva recorded at baseline (blue), PTZ TP1 (yellow) and PTZ TP2 (orange) at 3 dpf (left, $N = 473$), 5 dpf (center, $N = 547$) and 7 dpf (right, $N = 546$). PTZ significantly decreased inter-eye distance at 3 dpf (first TP) or 5 and 7 dpf (TP1 and TP2). Wilcoxon signed rank $*p < 0.05$, $****p < 0.0001$.

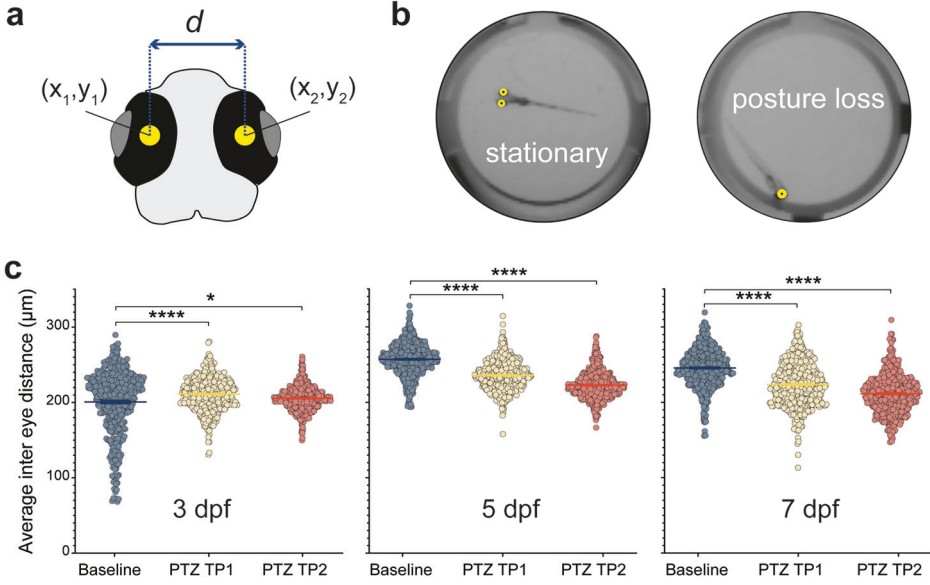

**Fig. 6 | Seizure activity reflected in inter-eye distance and brain activity. a** Representative inter-eye distance plotted over time for baseline and PTZ TP2 recordings (top) with a seizure event highlighted at higher resolution (bottom). **b** Outline of a zebrafish overlaid with an image of the *HuC:H2B:GCaMP6s* reporter expression. Live microscope image of photometry fiber placement with blue shading overlaid to indicate casted light from the probe. Placement of electrode is also shown for simultaneous LFP recording. **c** Representative baseline recording showing $Ca^{2+}$ trace recorded from probe aligned with LFP showing similar activity. 15 mM PTZ recordings from a larva showing $Ca^{2+}$ trace recorded from probe aligned with LFP showing seizure-like activity in the probe aligning with events LFP. Scale bar represents 50 μm.

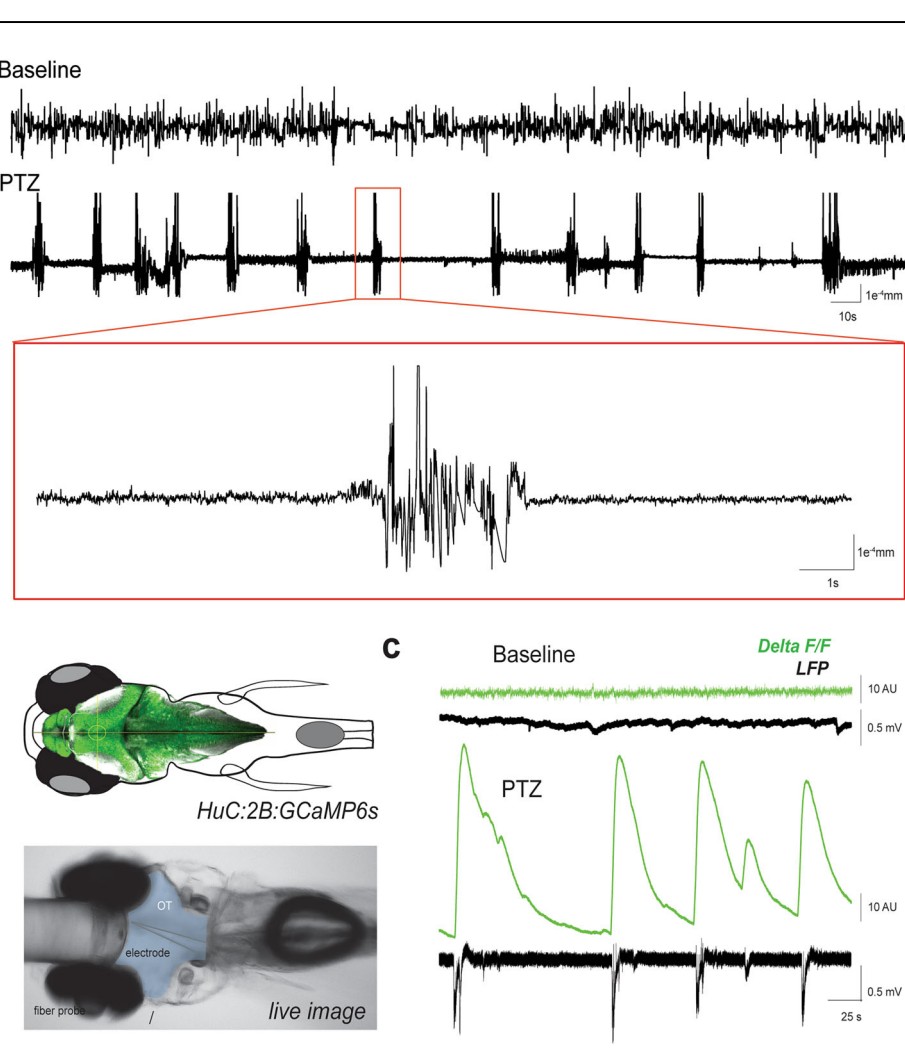

**Fig. 7 | Anti-epileptic effect of VPA.**
**a** Representative dual LFP (black) and fiber photometry Ca$^{2+}$ (green) recordings showing 15 mM PTZ induced seizure activity (above) followed by 5 mM VPA treatment (below) showing seizure ablation.
**b** Representative tracking plots for PTZ and 5 mM VPA treatments, respectively. **c** Plots for average total activity (A.U., left) and total distance (m, right) of each larva ($N = 768$) with PTZ (green) and 5 mM VPA treatment (green). Wilcoxon signed rank ****$p < 0.0001$.

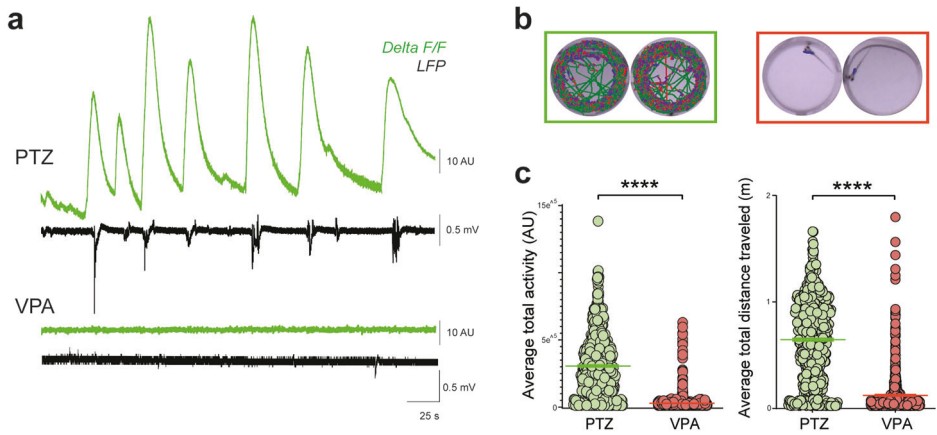

## Methods

### Zebrafish husbandry
We have complied with all relevant ethical regulations for animal use. All procedures described herein were performed in accordance with the Guide for the Care and Use of Animals (ebrary Inc., 2011) and adhered to guidelines approved by University of California, San Francisco Institution Animal Care and Use Committee (IACUC approval: #AN171512-03A). Adult and juvenile zebrafish were maintained in a temperature-controlled facility on a 14-h light and 10-h dark cycle (9:00 AM to 11:00 PM PST). Juveniles (30–60 dpf) were fed twice daily, once with JBL powder (JBL NovoTom Artema) and the other with JBL powder mixed with live brine shrimp (Argent Aquaculture). Adults were also fed two times per day, first with flake food (tropical flakes, Tetramin) and then with flake food mixed with live brine shrimp. Zebrafish embryos and larvae were raised in an incubator kept at 28 °C on the same light/dark cycle as the facility in embryo medium consisting of 0.03% Instant Ocean (Aquariam Systems, Inc) in reverse osmosis-distilled water. For PTZ behavioral experiments, all larvae were wild type AB strain (WT), and experimental time points were 3, 5 and 7 dpf. For behavioral experiments in genetic models, larvae were wild type sibling or homozygote mutants from F10 or later generation adult heterozygote (*gabrb3*, *pcdh19*, *pnpo*, *scn1lab* (s552), *scn8aa* or *syngap1b*) crosses and genotyped *post hoc*. Since zebrafish larvae do not undergo sexual differentiation until 20-25 dpf, sex is not an impacting biological factor on any result presented in this study.

### Experimental design, body orientation estimation and acquisition
Individual larvae were randomly selected and pipetted into wells of a 96-well plate in 150 μL embryo media and placed in the multi-camera array microscope (MCAM$^{TM}$, Ramona Optics Inc., Durham, NC, USA) to habituate for 20 min. For PTZ experiments, WT larvae were used; for genetic models (conspecific) WT siblings were used. The baseline period was acquired for 5 min followed by the addition of 50 μL PTZ directly into each well to reach a final concentration of 15 mM. After a 5 min incubation period, timepoint one (TP1) was acquired for 5 min followed by a 5 min wait period and acquisition of timepoint two (TP2) for 5 min. Imaging parameters for each 5 min recording epoch were set on a Linux workstation running custom Ramona Optics MCAM software. Specifications included 2 msec exposure, 2.0 digital gain, 1.25 analog gain, infrared (850 nm) transmission illumination with 65% brightness, 160 frames per second acquisition, and sensor pixel binning mode 4 to optimize frame rate to obtain a video of shape 256 × 256 pixels for each of the 96 wells. The height of the sample platform was adjusted for optimal focus of the larvae for each experiment using the MCAM interface for visual guidance. Raw image data was stored in NetCDF format with all relevant metadata.

Once all video data was acquired, they were compressed to MP4 format and tracked using algorithms built into custom Ramona Optics

MCAM software. Pose estimation machine learning models were originally based on the DeepLabCut backbone[84] and then optimized for parallelized inference and MCAM compatibility. Models were trained internally at Ramona Optics with a large amount of training data from many diverse datasets and fine-tuned with one thousand frames from datasets collected in this work specifically. The optimized body orientation estimation model (version 20230825) was evaluated for accuracy by comparing inferred key point locations to 1177 frames of manually annotated ground truth zebrafish and all larvae in all video recordings were tracked using this model (Supplementary Fig. 2). Tracking data provided cartesian coordinates of eight key-points on each zebrafish image, yielding 48,000 frames of data per recording epoch. Data acquisition continued until a minimum of 3 independent breeding replicates were performed for each age group and the following sample sizes were reached as a result: 473 recorded larvae at 3 dpf, 547 recorded larvae at 5 dpf, and 546 recorded larvae 7 dpf, followed by an additional 768 recorded larvae treated with VPA at 5 dpf. The same approach was followed for genetic mutant populations: $n = 74$, 101 (*gabrb3*, WT), $n = 89$, 74 (*pcdh19*, WT), $n = 79$, 147 (*pnpo*, WT), $n = 226$, 97 (*scn1lab*, WT), $n = 62$, 83 (*scn8aa*, WT), and $n = 136$, 45 (*syngap1b*, WT).

### Fiber photometry and electrophysiology
5 dpf *Tg(HUC:H2B-GCaMP6s)* larvae were immobilized dorsal side up in 2% low melting point agarose (BP1260-100, Fisher Scientific). Recording chambers were bathed in embryo media, placed on the stage of an upright microscope (Olympus BX-51W) and monitored continuously using a Zeiss Axiocam digital camera. Under visual guidance, a 100 μm photometry fiber (RWD Life Science Co., LTD) was placed against the forebrain to capture changes in Ca$^{2+}$ signals prior to placing a single-glass microelectrode (WPI glass #TW150 F-3) with approximately 1 μm tip diameter backfilled with 2 mM NaCl internal solution in the optic tectum for gap-free local field potential (LFP) recordings. Fiber photometry acquisition settings included a 60 frames per second capture rate, the gain value set to 70 and light settings set to 80% and 10% for 410 nm and 470 nm respectively. Fiber photometry data was stored on a computer running Multichannel Fiber Photometry Software (RWD Life Science Co,. LTD). LFP acquisition settings were conducted as described previously described[22,72] including low-pass filtering at 1 kHz, high-pass filtering at 0.2 Hz and sampling at 10 kHz using a Digidata 1320 A/D interface (Molecular Devices). LFP data was stored on a computer running AxoScope 10.3 software (Molecular Devices). Baseline recordings were taken prior to bath application of 15 mM PTZ (60 min exposure, $N = 6$ larvae).

### Pharmacology
Pentylenetetrazole (PTZ) (Sigma Aldrich, CAS: 54-95-5) was dissolved in embryo medium at a concentration of 60 mM and was used to chemically induce acute seizure activity at a final concentration of 15 mM[22].

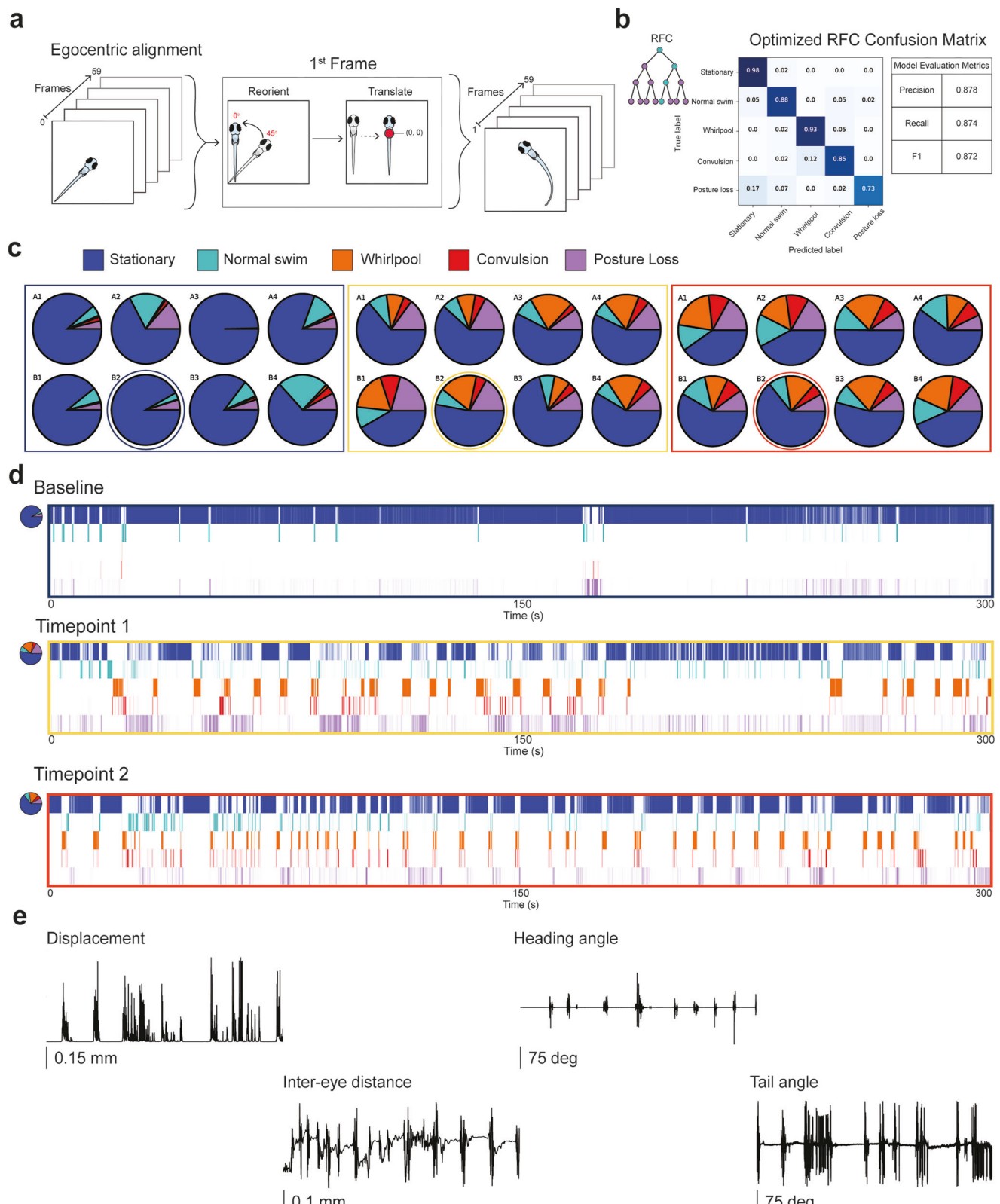

**Fig. 8 | Automated behavior classification with random forest classifier.**
**a** Schematic outlining process for egocentric alignment transformations and
translations for 60 frame windows. **b** Random Forest Classifier (RFC) model and
confusion matrix showing performance of the optimized model. **c** Sample pie charts
depicting percentage of each behavior performed for 8 individual larvae at each
recording epoch. Baseline (navy box), TP1 (yellow box), TP2 (orange box).
Represented behaviors include stationary (indigo), normal swim (teal), whirlpool

(orange), convulsion (red) and posture loss (purple). **d** Ethogram plot (top) over the
entire duration of each recording epoch for one representative larva from well B2
(circled in panel (**e**), TP2, scalar measurements plotted that align with ethogram
showing how various behaviors appear when represented as larval displacement
(mm), inter-eye distance (mm), change in heading angle (deg), or tail angle (deg)
over time (bottom).

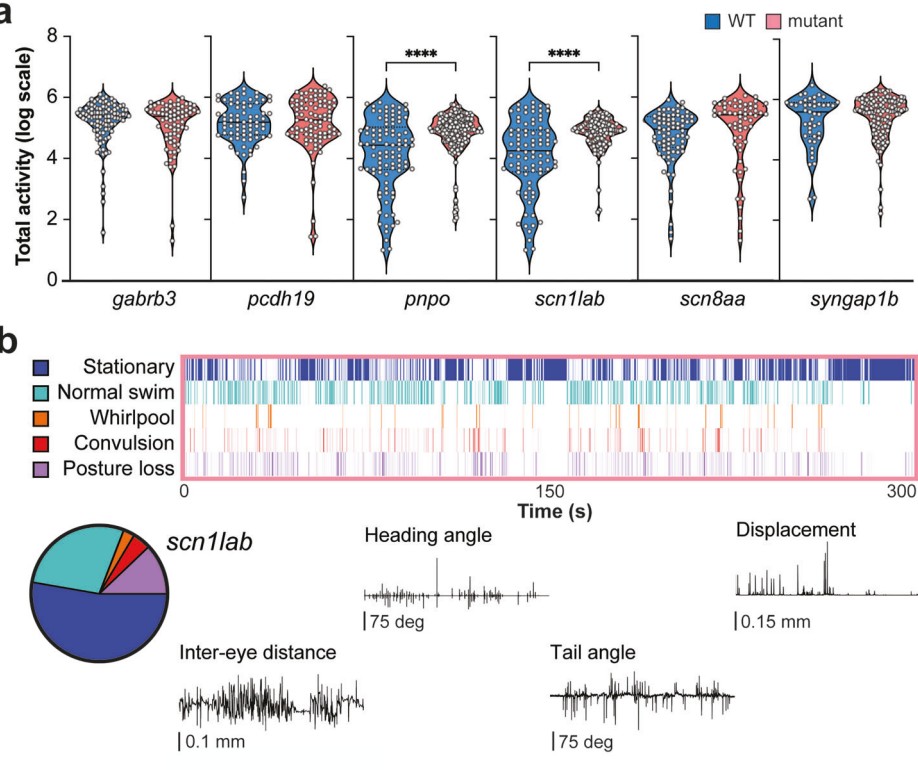

**Fig. 9 | Automated MCAM analysis on genetic epilepsy models. a** Total activity dot-plots are shown for six CRISPR-generated zebrafish congenic WT siblings and age-matched homozygote mutant lines: *gabrb3* (101 WT, 74 mutant); *pcdh19* (74 WT, 89 mutant); *pnpo* (94 WT, 184 mutant; unpaired t-test ****$p < 0.0001$); *scn8aa* (79 WT, 60 mutant); *scn1lab* (84 WT, 137 mutant; unpaired t-test ****$p < 0.0001$); *syngap1b* (45 WT, 133 mutant). **b** Ethogram plot for a representative *scn1lab* homozygote mutant zebrafish. The entire duration of the 5 min recording epoch for one representative larva is shown (top). Pie chart depicting percentage of each behavior performed at each recording epoch (bottom, left). Color coding as in Fig. 8. Scalar measurements plotted that align with ethogram showing various seizure-like behaviors (bottom, right).

## Activity metric data processing

The total activity metric ($A_i$) was computed for each fish by summing the change in pixels between sequential frames of an acquired video according to Eq. (1). In this equation $P_i(x,y)$ is the pixel value of frame i, $P_{i+1}(x,y)$ is the pixel value of frame i + 1, T is a relative threshold set to 0.1, D is an absolute threshold set to 20, m is the frame height, and n is the frame width. This endpoint was used to determine gross movement. Larvae with total activity metric scores below 20,000 pixels during TP2 were automatically excluded from pose estimation analysis as this activity threshold was established for larvae that did not survive PTZ treatment.

$$A_i = \sum_{x=1}^{n} \sum_{y=1}^{m} \left[ \frac{2 * |P_i(x,y) - P_{i-1}(x,y)|}{P_i(x,y) + P_{i-1}(x,y)} > T \right] \bigcap [|P_i(x,y) - P_{i-1}(x,y)| > D]$$ (1)

## Data cleaning

All scalar data, including both *x* and *y* coordinates for each of the 8 key-points on the zebrafish larvae, was filtered and smoothed according to the following procedure prior to data analysis. Larval speeds captured in this dataset, reaching 143 million data points, were plotted as a histogram to establish a biologically relevant speed threshold of 120 mm/sec and verified further by eye. Frames where larvae were moving faster than this threshold were considered anomalies and the erroneous frames were excluded from the tracking data. Frames were also removed if the center key-point of the larvae jumped outside the boundary of the circular well or if any key-point was found to be over 0.7x the larval length away from the center of mass (COM) of the larva. COM for each larva was calculated from 6 points along the skeleton midline as the 2 eye key-points biased the COM to the front. Data was then smoothed using a wavelet denoising filter (Daubechies 1990) implemented with Scikit Image (scikit-image 0.21) and the sym4 wavelet (scikitlearn 1.2)[101,102] see Eq. (2). Sigma values for use with the denoising filter were computed as the average sigma for each feature of each well plate acquisition. Once data was denoised, the following scalar metrics were analyzed: total distance traveled, instantaneous speed, changes in tail and

heading angles, and inter-eye distance.

$$CWT_\psi f(a, b) = W_f(b, a) = |a|^{-\frac{1}{2}} \int_{-\infty}^{\infty} f(t)\psi^* \left( \frac{t - b}{a} \right) dt$$ (2)

## Machine learning model training

Video frames were organized into overlapping vector windows of 60 frames, corresponding to 0.375 s per window[53]. The 60 frame vector windows were extracted as video clips and annotated as one of five behavior classes: stationary, normal swim, whirlpool, convulsion and posture loss. A dataset of 4418 clips were labeled and for each behavior class 90% were randomly selected as the training set and 10% as the test set. To balance the representation of each behavior class in the test set for model evaluation, 409 samples were selected from each class's test set. 409 was the number of samples chosen as this was the number of prelabeled samples available for the behavior class that occurred the least. Any remaining test sets from the other class's were added back to the training set.

Egocentric alignment transformations of estimated poses were computed for the first frame of each window, which rotates the fish within the well to a vertical (head up) position followed by a translation that shifts all key-points to the center of each image. This series of transformations were applied to the remaining 59 frames in each vector. Each aligned feature vector of shape 60 × 8 × 2 (frames x key-points x coordinates) was flattened to a feature vector of length 960 for further use.

Feature vectors corresponding to video clips were normalized by subtracting the mean value and dividing by the maximum value of each feature. Principal component analysis (PCA) was used to reduce dimensionality (https://sciwheel.com/work/bibliography/16334125) (scikit-learn 1.3.2), retaining components describing 95% of the dataset variance. Normalization and PCA were computed using only the training set, then the same transformations were applied to the test set to ensure that the estimated accuracy of the algorithm was not contaminated. The normalized and aligned vectors of key point data were input to the model, which was trained to fit the data space. These feature vectors inherently capture the

behavioral kinematics we computed, eliminating the need to input them as separate features.

Three different algorithms were evaluated for behavioral prediction of video clips (scikit-learn 1.3.2): k-nearest neighbors[103], random forest[104], and support vector classifiers[105]. All models were trained for one thousand iterations. Precision, recall and F-score were averaged across all classes and were used as evaluation metrics[106] for comparison according to Eqs. (3) and (4), where TP is the True Positive rate, FP is the False Positive rate and FN is the False Negative rate. Confusion matrices were generated for visualization of model accuracy and F1 score was evaluated for each model according to Eq. (5), where $M_{ij}$ is the confusion matrix element at index $i, j$ and $n$ is the number of classes and thus dimension of the confusion matrix. The random forest algorithm was chosen for further optimization and trained, allowing dataset shuffling between training iterations, resulting in tens of thousands of different models being trained. The best performing random forest model was selected for final use in Fig. 8.

$$Precision = \frac{TP}{TP + FP} \tag{3}$$

$$Recall = \frac{TP}{TP + FN} \tag{4}$$

$$F1 = \frac{\sum_i M_{ii}}{\sum_{i=1}^{n} M_{ii} + \frac{1}{2}\left[\left(\sum_{i=1}^{n}\sum_{j=1, i\neq j}^{n} M_{ij}\right) + \left(\sum_{i=1}^{n}\sum_{j=1, i\neq j}^{n} M_{ji}\right)\right]} \tag{5}$$

All pose estimation tracking data was organized using the same windowing and egocentric alignment described above. The random forest model was used to predict behavioral states for individual larvae and results were visualized by generating ethograms and aligning them with scalar metric tracking results. For simplistic behavioral comparisons across larvae, pie charts demonstrating the calculated fraction of time spent for each behavior during a recording epoch were generated for each larva.

## Statistics and reproducibility

Statistical analysis was conducted using GraphPad Prism 10.0.2 software. Data distributions were assessed for normality using both the Shapiro–Wilk and Kolmogorov–Smirnov tests. Both tests indicated that all PTZ and VPA data were not normally distributed, therefore, non-parametric statistical testing was applied. Specifically, the Wilcoxon matched-pairs signed-rank test was used to determine whether population mean ranks differed. Data from the genetic mutant models were normally distributed and an unpaired t-test was used to determine significance. An alpha level of 0.05 was used for all comparisons and all p-values were two-tailed. Statistical significance was defined as $p < 0.05$, unless otherwise noted. Final sample sizes after outlier removal were $n = 473$ (3 dpf), $n = 547$ (5 dpf), and $n = 546$ (7 dpf).

## Reporting summary

Further information on research design is available in the Nature Portfolio Reporting Summary linked to this article.

## Data availability

Examples of raw data collected with this project in addition to video samples of various larval behavioral seizure activities are available in three data repositories which can be found at: https://zenodo.org/records/15352042[107], 10624845[108], 10625177[109], or 10565088[110]. The complete data has been retained locally and can be provided upon reasonable request.

## Code availability

Code related to data cleaning and calculating distances, heading and tail angles, as well as inter-eye distances are written in python and are available

in a Git repository. Supervised machine learning algorithms, along with ethogram generation, are also located in the repository which can be found at: https://gitlab.com/ramona-applications/seizure_behavior_analysis.

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

## Acknowledgements
This work was supported by National Institutes of Health grants R01-NS096976, R01-HD102071, and U54-NS117170 (to S.C.B.). This work was also supported by funding issued to Ramona Optics by the Office of Research Infrastructure Programs (ORIP), Office of the Director, National Institute of Environmental Health Sciences (NIEHS) and the National Institutes of Health grant R44OD024879. The National Cancer Institute (NCI) grant R44CA250877, the National Institute of Biomedical Imaging and Bioengineering (NIBIB) grant R43EB030979, the National Science Foundation grant 2036439, and the National Institute of Mental Health grant R43MH133521.

## Author contributions
Conceptualization, P.W.-F. and S.C.B.; methodology, P.W.-F., J.E., and M.H.; experimentation and data acquisition, A.V., A.C. and P.W.-F.; computational analysis, J.E. and P.W.-F.; software J.E., S.C., T.J.J.D., M.H.; visualization, P.W.-F., J.E, A.B., M.H., and S.C.B.; funding acquisition, S.C.B. and M.H.; project supervision, S.C.B., M.H., A.B. and P.W.-F.; writing – original draft, P.W.-F., J.E, M.H. and S.C.B.; writing – revised draft, S.C.B. with review and editing from J.E. and P.W-F.

## Competing interests
The authors declare the following financial and personal relationships that may be considered as potential competing interests: S.C.B. is a consultant for Harmony Biosciences. J.E., A.B., T.J.J.D and M.H. have a financial interest in Ramona Optics Inc. The remaining authors declare that the research was conducted in the absence of any commercial or financial relationships that could be construed as a potential conflict of interest.
