## [Transparent Peer Review file · Communications Biology]

Automated detection of complex zebrafish seizure behavior at scale

Corresponding Author: Professor Scott Baraban

Version 0:

Reviewer comments:

Reviewer #1

(Remarks to the Author)

The manuscript by Whyte-Fagundes et al., presents an interesting novel approach for the automation of seizure detection in larval zebrafish. The use of a high-resolution multi-camera system combined with supervised machine learning (ML) algorithms for analyzing movements and kinematic parameters provides a promising model for studying epileptic seizures in a small animal model, such as zebrafish. Notably, the ability to perform eight-point skeletal pose tracking and analyze behaviors in a high-density 96-well format represents a significant advancement for large-scale studies. While the framework is innovative and represents notable progress in automated seizure detection, several critical limitations need to be addressed before publication. By addressing these major revisions, the study would significantly enhance its translational impact and robustness, thereby solidifying its contribution to the field of epilepsy research.

Major revisions:

- 1) The study exclusively uses pentylenetetrazole (PTZ) to induce seizures, a compound that provokes extreme and artificial seizure-like behaviors. This limits the model's applicability to real-world epileptic conditions. The inclusion of models with spontaneous seizures, such as genetic or idiopathic epilepsy models, is essential to improve validity of the system.
- 2) To demonstrate that the system is suitable for drug screening, experiments involving the administration of antiepileptic drugs should be included. Testing the effects of these compounds on behavior would validate the system's ability to detect changes induced by pharmacological interventions, an essential aspect for screening potential therapeutic agents.
- 3) A final short paragraph on the limitations of the study should be included

Reviewer #2

(Remarks to the Author)

Review of Manuscript Number: COMMSBIO-24-7986-T

Title: Automated detection of complex zebrafish seizure behavior at scale

Overall

This is a nicely written and interesting paper tackling an important subject with a well executed study.

The work does push the field forwards a little, but I feel that the study is a bit of an opportunity missed by not looking at more subtle markers and incorporating these into a ML-based approach for automatic detection of convulsions. After all, not all seizures/convulsions are like those induced by PTZ exposures.

Consequently, I would have hoped for a wider study demonstrating the broader utility of this approach for detecting different types of seizure. For example, how does the model behave with other mechanistic classes? GABAA antagonist exposure produces characteristic and strong tonic-clonic like convulsions as you know, but other mechanisms generate different behavioural profiles (e.g. AChE inhibitors). The important next step is to identify defining features of other classes which will

also provide information on the likely mechanism of actions of antiepileptic drugs and also facilitate the identification of convulsions after exposure to new drugs - this would be a powerful tool for use in drug discovery and feel the study would have benefitted from a little more content and depth in this respect. Having said this, if the aim is to publish as a method, then it does the job, but other things such as a true test of predictive value with 'unknowns' could have been added with little extra effort.

Also do you not feel the use of 96 well plates restricted the potential resolution of your approach? Looking at the SI videos gave me the impression that more subtle behaviours, especially associated with posture, could be gleaned by provided a larger tracking area in which the animals would not bounce off/around the edges of the wells constantly. Also in 30501_0_video_9925169_sn6b27.mp4 I thought a measure of gait would be relevant here – the tail movement reminds me of rodent loss of gait so finding some way of capturing this would be great.

Abstract

I found the abstract a little abrupt and generic but perhaps that is just journal style which I am not too familiar with. For example, I am not sure it is entirely accurate to state that you “evaluated seizure behavior in over 2400 freely swimming larval zebrafish exposed to pentylentetrazole (PTZ), a chemoconvulsant”. My understanding is that you analysed these to establish the characteristics of a convulsion rather than assessed these post model establishment and if so this should be clarified.

Line 20 - Convulsive seizure-associated behaviors: I would prefer the measured behavioural response to be referred to in this manner throughout as there is no concomitant measure of electrographic impacts to truly indicate the occurrence of a seizure in parallel with these behaviours (although implied by your ephys/Ca²⁺ data).

Line 23 - automated detection of larval zebrafish convulsions

Line 24 – seizure-associated behaviour or convulsions, either is acceptable given these are behavioural manifestations of electrographic seizures

Line 24 – I presume you mean you analysed 2400 larvae to train your model and identify? – Please be more specific

Line 26 — I would prefer to refer to this as body pose or orientation as your key points are on the body rather than the skeleton.

Introduction

Line 37 – any seizures in any species, even in isolation, can result in behavioural outcomes as convulsions. Suggest remove Epileptic human. Also are these complex behaviours – stereotypical or characteristic, but not really complex, like learning or sociality for example.

Line 41 – I find the reference to natural models of epilepsy a little odd without further explanation – I would remove this reference and also expand the introduction to include studies that support the use of zebrafish for seizure analysis from a biological perspective, as most of the points made are about convenience. For example, what about comparable brain circuitry, similar ECG characteristics etc – these are important to state to support the approach, especially given this is a general biology journal and most readers will be not familiar with the translational value of the model.

Line 63 - dissolved in bathing medium can be rapidly absorbed directly through the skin. This is not the case for all compounds, many antibiotics being an example.

Line 69 – I think the real advantage is to improve the accuracy of detecting seizures and the effects of AEDs rather than offering greater insight into epilepsy per se – I am not sure how that would be the case? I think make this clearer in this sentence and also that this applied to zebrafish specifically.

Line 73/74 – Again, I am not sure any of the endpoints used constitute complex behaviours.

Line 81 – did speeds reach 120mm.sec – i.e. were the actual speeds measured, or did they exceed 120 mm/sec? Please be more precise

Line 85 – I feel like the introduction needs a final line to convey the wider impact of this model, for example how it will aid high throughput assessment of AEDS and potentially proconvulsive compounds.

Results

Line 104 - Accuracy evaluation of the pose estimation model suggested inferred coordinate errors ranging from 55 to 130 microns depending on the 106 key-point coordinate – what does this actually mean in real terms for model performance? Can you be more explicit as 130 microns is a considerable portion of a larval zebrafish

Line 118-158 – important data – I would like to see a brief description here of how the 3 age groups differed from one another in their behavioural profiles before and after PTZ application – the description is (understandably) focused on demonstrating a similar response between the 3 but the differences are important for labs using different stages for various reasons

Line 157 – these are all convulsive behaviours are they not – just at different stages – just refer to different stages of convulsive behaviour based on your 2005 paper.

Line 160 – the kinematic data are key here, and the really novel bit, especially when analysing different types of convulsive behaviours, identifying more subtle impacts of pro and anticonvulsive compounds etc. For me, these need to be incorporated into the ML approach (see below) – they may not be revealing with 15mM PTZ (and perhaps were removed after PCA?), but my guess is that they will become much more important once you start to look at more subtle mechanisms or lower concentrations.

Line 199 onwards – refer to as Ca²⁺ rather than calcium.

Line 201 – Although the Ca²⁺ imaging is elegantly performed, especially with simultaneous LFP recording, I struggle to see how this relates to the abrupt changes in inter-eye distance you describe since you cannot observe this as occurring during a Ca²⁺/ECG spike? Can you make it more clear how this helps.

Also are the ‘twists and turns’ not just a product of animal bouncing off the sides of the well during a convulsion? This also has implications for swim speed – my guess in an open arena swim speeds will be higher than those you state, and this goes back to my thought that larger area would provide more subtle measures of a convulsion.

Often this loss of balance is observed during inter-ictal periods – did you analyse baseline versus treatment period frequency of this as I suspect it happens a lot more during interictal periods than during baseline (or at least the occurrence of this loss of posture as a proportion of time spent not moving)

Lines 224-228 – Why were more kinematic endpoints not included in the model in some capacity? I suspect these more subtle and finer scale measures will become more important once you look beyond GABAA antagonism as an initiating mechanism.

Did you assess for dose dependency and if so, what was the threshold for detection against baseline?

How about the ability of automated detection of seizures in plate of randomly distributed treated and untreated larvae? What was the detection rate, FPs and FNs?

Discussion

Line 267 – did they enable automated detection of complex seizure behaviors in larval zebrafish at scale? I didn't see application of this, for example by testing in a plate of randomly distributed treated and untreated larvae. If not, this should be rephrased as the possibility to automatically detect – better still test it and include the data – it's a straightforward and quick experiment.

Line 287 – “constrained by smaller arenas or single-subject tracking” Not sure any of these studies used areas smaller than a 96 plate well? – Please clarify - In my opinion a 96 plate well is already too small as discussed above.

296 – As stated above, I would like to see incorporation of the kinematic endpoints into the ML approach for detection of more subtle events - this needs to be discussed here along with the fact that only one compound/mechanism was tested, no concentration responsiveness was shown, and also an acknowledgement that the assessment of PTZ-induced convulsions is likely to be at the more severe end of the scale should be made

Methods

Line 331 – was the tracking undertaken in the light or dark?

Line 334 – was thorough mixing achieved?

The final concentration of PTZ used was high - others have demonstrated behavioural effects at much lower concentrations – why was this concentration selected and did you assess lower concentrations to establish a threshold for detection?

Line 363 – was no chemical used for restraint alongside the agarose?

Line 367 – Ca²⁺ rather than calcium

Line 363 – what was the contents of the LFP recording chamber?

Line 405 – “.....considered anomalies and the erroneous frames were excluded from the tracking data”. Why were these considered anomalies – was there a statistical threshold you used to objectively discard these data points?

Line 407 – what was the justification for using 0.7x the larval length away from the center of mass (COM) of the larva as the cut off point here?

Figures

The figures are very nice and very informative

Reviewer #4

(Remarks to the Author)

In their manuscript titled "Automated detection of complex zebrafish seizure behavior at scale," Whyte-Fagundes and colleagues employ a multi-camera imaging system enabling high-throughput behavioral assays to track larval zebrafish seizure activity. Additionally, they develop a machine-learning-based algorithm for the offline, automated detection of both normal and abnormal behaviors.

The study is well-designed and potentially valuable for a broad range of researchers using zebrafish as a model organism, beyond epilepsy research alone. However, I have concerns that should be addressed before recommending the manuscript for publication in *Communications Biology*.

Major concerns:

- 1- The authors should provide a rationale for using 5-minute tracking intervals rather than continuous recording when monitoring motor seizures.
- 2- How do authors explain the reduction in locomotor activity and total distance traveled between 5 and 7 dpf larvae (Fig. 2a and 2b)?
- 3- The manuscript refers to a representative plot for inter-eye distance as Figure 5d (lines 190-191), but panel d is not present in Figure 5. Are the authors instead referring to Figure 6a, or is a figure panel missing?
- 4- In lines 195–197, the authors claim: "It is not feasible to simultaneously record inter-eye distance and abnormal brain activity in seizing larvae moving at speeds above mm/sec." This statement is quite strong. Do the authors have evidence to support this? It seems premature to exclude the possibility of achieving this goal using wide-field calcium imaging, light-field microscopy, or tracking microscopy. For context, I encourage the authors to consider the following references:
PMID: 28596596
PMID: 32778840
PMID: 28892088
- 5- The manuscript notes similarities between baseline/PTZ inter-eye distance traces and those of LFP + photometry (lines 207-209). However, the parallelism is not fully explained. For example, similarities could also exist between brain activity recordings and locomotor activity variables (e.g., distance traveled or heading angle). The authors should clarify this parallelism to ensure it does not come across as a trivial observation (e.g., both traces merely showing episodes of activity interspersed with baseline states).
- 6- The authors claim that machine learning-based detection of motor seizure patterns can serve as a surrogate measure for brain activity (lines 300-302). While every movement undeniably reflects underlying neuronal activation, the inverse is not necessarily true. This point, which links to the previous one, requires clarification.
- 7- Why was a zebrafish line with nuclear-localized calcium reporters chosen instead of one with cytoplasmic localization, given that fiber photometry captures aggregate, non-cellular resolution measurements? Please provide rationale for this choice.

Minor:

- Statistical analysis: please include reference to the use of ANOVA statistics employed in Figure 2 and 5.
- Figure 1b: color code for larval speed should be included, as provided in Supplementary Figure 1e.
- Line 188: The sentence should reference Fig. 5c.
- Figure 6a: Given the small inter-eye distance, I suggest reporting this measurement in microns instead of millimeters.

Version 1:

Reviewer comments:

Reviewer #1

(Remarks to the Author)

The Authors have significantly improved their text and the structure of the manuscript is now more sound. The development of this new tool for the investigation on a large scale of "seizure-like" events and the possible role of ASM in models of DEE in zebrafish represent an important advancement in preclinical studies of DEE

Reviewer #2

(Remarks to the Author)

The authors have addressed all of the comments sufficiently and have greatly improved the manuscript, especially through the addition of extra experimental data and addressing some of the limitations more explicitly. I am very happy to accept

publication.

Reviewer #4

(Remarks to the Author)

The authors have addressed satisfactorily all of my comments. Considering the overall revision work carried out, I endorse the manuscript for publication in Communications Biology.

Point-by-point response to 50 comments from 4 reviewers are shown below in BLUE

Reviewer #1 (Remarks to the Author):

The manuscript by Whyte-Fagundes et al., presents an interesting novel approach for the automation of seizure detection in larval zebrafish. The use of a high-resolution multi-camera system combined with supervised machine learning (ML) algorithms for analyzing movements and kinematic parameters provides a promising model for studying epileptic seizures in a small animal model, such as zebrafish. Notably, the ability to perform eight-point skeletal pose tracking and analyze behaviors in a high-density 96-well format represents a significant advancement for large-scale studies. While the framework is innovative and represents notable progress in automated seizure detection, several critical limitations need to be addressed before publication. By addressing these major revisions, the study would significantly enhance its translational impact and robustness, thereby solidifying its contribution to the field of epilepsy research.

Major revisions:

1) The study exclusively uses pentylenetetrazole (PTZ) to induce seizures, a compound that provokes extreme and artificial seizure-like behaviors. This limits the model's applicability to real-world epileptic conditions. The inclusion of models with spontaneous seizures, such as genetic or idiopathic epilepsy models, is essential to improve validity of the system.

We agree with this excellent suggestion. As the PTZ model is the most established and widely used larval zebrafish epilepsy model it is a logical choice to develop the MCAM system based on this model. That said, the substantially revised manuscript now includes new validation experiments using six different stable genetic zebrafish models representing a range of developmental epileptic encephalopathies (DEEs) generated using CRISPR technology in our laboratory (Griffin et al. 2021), with a detailed focus on the *scn1lab* zebrafish model representing Dravet syndrome e.g., one of the most severe DEEs and best characterized zebrafish model exhibiting spontaneous seizures.

New text added (lines 293-314). New Figure 9 added.

2) To demonstrate that the system is suitable for drug screening, experiments involving the administration of antiepileptic drugs should be included. Testing the effects of these compounds on behavior would validate the system's ability to detect changes induced by pharmacological interventions, an essential aspect for screening potential therapeutic agents.

We also agree with this astute suggestion. One purpose of advancing the MCAM system using a 96-well plate format is to provide a next-generation iteration of high-throughput drug screening for antiepileptic drugs using larval zebrafish. However, full high-throughput drug screening projects testing 100s to 1000s of drugs can take several years or more to complete. That said, the revised manuscript now includes proof-of-principle studies with valproate acid (VPA), a commonly used antiepileptic drug.

New text added (lines 237-250). New Figure 7 added.

3) A final short paragraph on the limitations of the study should be included

A significantly revised final paragraph (lines 384-402) now includes a discussion of limitations. Specifically, the following text was added: “However, the high-resolution acquisitions required to perform these analyses on a 96-well plate are limited by computing power currently available. For example, current hardware presents physical limitations related to the speed at which central processing unit (CPU) process instructions, the amount of memory available and the size of storage devices which resulted in the relatively brief 5-minute acquisition epochs used here. Since zebrafish move in three dimensions and images capture only two, the current system may not fully represent the space occupied by a seizing larva.”

Reviewer #2 (Remarks to the Author):

Overall

This is a nicely written and interesting paper tackling an important subject with a well-executed study.

4) The work does push the field forwards a little, but I feel that the study is a bit of an opportunity missed by not looking at more subtle markers and incorporating these into a ML-based approach for automatic detection of convulsions. After all, not all seizures/convulsions are like those induced by PTZ exposures.

First, using the higher resolution imaging possible with the MCAM system we did include new “subtle” markers of seizure behavior not previously reported in the literature e.g., activity, tail angle and inter-eye distance metrics. Second, we now include additional new experiments (text lines 293-314; Figure 9) to apply this ML-based approach to 6 genetic models of epilepsy with less severe seizure behavior phenotypes than those associated with bath exposure to a convulsant, PTZ. Subtle differences between acute PTZ seizure behavior and spontaneous seizure behavior seen in a zebrafish model for Dravet syndrome (*scn1lab*) are noted. Specifically, convulsive seizure-like behaviors in mutant fish are briefer and do not include a post-ictal like loss of posture.

5) Consequently, I would have hoped for a wider study demonstrating the broader utility of this approach for detecting different types of seizure. For example, how does the model behave with other mechanistic classes? GABAA antagonist exposure produces characteristic and strong tonic-clonic like convulsions as you know, but other mechanisms generate different behavioural profiles (e.g. AChE inhibitors). The important next step is to identify defining features of other classes which will also provide information on the likely mechanism of actions of antiepileptic drugs and also facilitate the identification of convulsions after exposure to new drugs - this would be a powerful tool for use in drug discovery and feel the study would have benefitted from a little more content and depth in this respect. Having said this, if the aim is to publish as a method, then it does the job, but other things such as a true test of predictive value with ‘unknowns’ could have been added with little extra effort.

We agree that the MCAM system can and should be applied more broadly to other larval zebrafish epilepsy models. As noted above, new studies have been added to the revised manuscript with additional genetic models (see new Figure 9). Further, how an antiepileptic drug impacts seizure behavior (and brain activity) is now shown here for VPA (see new Figure 7). Unfortunately, defining the precise mechanism of action is something not definitively established for any existing FDA-approved antiseizure medication (ASM). In preclinical models, whether larval zebrafish or rodents, ASMs commonly are shown to simply reduce the frequency or occurrence of observable seizure behavior and this information does little to inform

underlying MOA. That said, we do believe the MCAM system is a next-generation advancement for large-scale screening and identification of new drugs but studies on MOA here are far beyond the scope of the current manuscript.

6) Also do you not feel the use of 96 well plates restricted the potential resolution of your approach? Looking at the SI videos gave me the impression that more subtle behaviours, especially associated with posture, could be gleaned by provided a larger tracking area in which the animals would not bounce off/around the edges of the wells constantly. Also in 30501_0_video_9925169_sn6b27.mp4 I thought a measure of gait would be relevant here – the tail movement reminds me of rodent loss of gait so finding some way of capturing this would be great.

On the contrary, the distinct advantage of larval zebrafish epilepsy models lies in the ability to perform single 96-well plate experiments at a scale not possible with rodent epilepsy models. This applies to phenotyping studies as shown here for 96 independent larvae, or high-throughput drug discovery programs. As such, demonstrating the capabilities of this high-resolution system using a 96-well plate format is precisely how we envision this approach being employed. In our experience, and in published studies, larval seizure behaviors described here in a 96-well format are similar to those reported in larger 6-, 24- and 48-well plates (PMID: 28812061; PMID: 24291671; PMID: 27895618).

We also considered what behavior could be interpreted as a larval zebrafish “gait” measurement akin to rodents but could not find a clear metric to capture this movement. As swim movements are not directly related to walking movements we are not comfortable relating these movements observed in distinctly different species as a fish measure of “gait”.

Abstract

7) I found the abstract a little abrupt and generic but perhaps that is just journal style which I am not too familiar with. For example, I am not sure it is entirely accurate to state that you “evaluated seizure behavior in over 2400 freely swimming larval zebrafish exposed to pentylenetetrazole (PTZ), a chemoconvulsant”. My understanding is that you analysed these to establish the characteristics of a convulsion rather than assessed these post model establishment and if so this should be clarified.

The Abstract confers to stated limitations (150 words or fewer) for *Communications Biology*. However, some edits were made (lines 24-26 and 30-31), as suggested.

8) Line 20 - Convulsive seizure-associated behaviors: I would prefer the measured behavioural response to be referred to in this manner throughout as there is no concomitant measure of electrographic impacts to truly indicate the occurrence of a seizure in parallel with these behaviours (although implied by your ephys/Ca²⁺ data).

Text was amended to be more measured in its conclusions.

9) Line 23 - automated detection of larval zebrafish convulsions

We kept this statement broad to generalize the study to the limitations seen across experimental animal models of epilepsy including rodents.

10) Line 24 – seizure-associated behaviour or convulsions, either is acceptable given these are behavioural manifestations of electrographic seizures

Edited to read “seizure-associated behavior”.

11) Line 24 – I presume you mean you analysed 2400 larvae to train your model and identify? – Please be more specific

Amended lines 24-26 as follows: “We assessed data from over 3,500 zebrafish either exposed to pentylenetetrazole (PTZ) or genetic zebrafish lines representing Developmental Epileptic Encephalopathy (DEE).”

12) Line 26 — I would prefer to refer to this as body pose or orientation as your key points are on the body rather than the skeleton.

Edited to read as body “pose”.

Introduction

13) Line 37 – any seizures in any species, even in isolation, can result in behavioural outcomes as convulsions. Suggest remove Epileptic human. Also are these complex behaviours – stereotypical or characteristic, but not really complex, like learning or sociality for example.

Term “epileptic human” was removed, as suggested. The term complex was replaced with stereotypical (line 38). Complex behaviors mentioned throughout are related to zebrafish larvae specifically.

14) Line 41 – I find the reference to natural models of epilepsy a little odd without further explanation – I would remove this reference and also expand the introduction to include studies that support the use of zebrafish for seizure analysis from a biological perspective, as most of the points made are about convenience. For example, what about comparable brain circuitry, similar ECG characteristics etc – these are important to state to support the approach, especially given this is a general biology journal and most readers will be not familiar with the translational value of the model.

Edits made to now read (lines 42-43): “These range from spontaneous models of epilepsy (dogs, baboons and domoic-acid poisoned California sea lions)”.

We would like to keep this statement to introduce the two types of epilepsy models as we include both types in our manuscript (i.e., PTZ being an acute model and genetically modified zebrafish larvae representing various DEEs being spontaneous).

15) We appreciate your suggestion to expand the Introduction to include a broader description of the translational value of the zebrafish model and included edits into the following paragraph that touch upon the conserved neuroarchitecture, similarities in cell types, LFP components, as well as genetic amenability and translational drug discovery.

Substantial edits and references added (lines 58-86) to address these concerns.

16) Line 63 - dissolved in bathing medium can be rapidly absorbed directly through the skin. This is not the case for all compounds, many antibiotics being an example.

Most published larval zebrafish drug applications are based on adding water-soluble compounds to the bathing medium (PMID #s 37215570, 22155488, 30711622, 29145140,

30500431, 26637551, 30917585, 33810553, 21147203, 17475807, 23000151, 19078942, 26688204, 34070577, 26477937, 29019661, 22197677, 34400286, 35134486, 31731399, 34075074, 25385118, & 34643222.). Here we only used water-soluble drugs for the pharmacology experiments. That said, at least two adult zebrafish studies did use bath application of antibiotics (PMID #s: 29608420 & 36625966).

17) Line 69 – I think the real advantage is to improve the accuracy of detecting seizures and the effects of AEDs rather than offering greater insight into epilepsy per se – I am not sure how that would be the case? I think make this clearer in this sentence and also that this applied to zebrafish specifically.

This entire paragraph (lines 58-83) was edited to focus on the accuracy of seizure detection and ASM drug discovery issues.

18) Line 73/74 – Again, I am not sure any of the endpoints used constitute complex behaviours.

The scientific community widely accepts zebrafish larvae as a model with complex behaviors and this is backed by numerous peer-reviewed studies (PMID #s: 23738739; 27019459; 30917585; 31866454 ; 34764296; 29561707; 22426213; 32518300 & 26502351). The measures we utilize here are standard and sensitive, capturing nuances in seizure-like patterns, which are recognized as complex in both larvae and adults.

19) Line 81 – did speeds reach 120mm.sec – i.e. were the actual speeds measured, or did they exceed 120 mm/sec? Please be more precise

Speed was measured. Edit in text for clarification (lines 96-97): “Abnormal high-speed convulsive behaviors were identified at maximum swim speeds of 120 mm/sec in a single well”.

20) Line 85 – I feel like the introduction needs a final line to convey the wider impact of this model, for example how it will aid high throughput assessment of AEDS and potentially proconvulsive compounds.

Final Introduction paragraph (lines 93-106) was edited. The last sentence (line 104) now reads: “Adapting these technologies to analyze clinically relevant seizure behaviors could significantly improve ASM evaluation by enabling more accurate, scalable behavioral screening and accelerating drug discovery.”

Results

21) Line 104 - Accuracy evaluation of the pose estimation model suggested inferred coordinate errors ranging from 55 to 130 microns depending on the 106 key-point coordinate – what does this actually mean in real terms for model performance? Can you be more explicit as 130 microns is a considerable portion of a larval zebrafish

Take any dataset, fish lengths are reported in a tracking output csv. Average fish lengths across the well plate, 3 to 3.5 mm is expected. Divide the minimum and maximum errors (55 and 130 microns) by the fish length and use these values as a reference. “The maximum error in the model is approximately 1/25th of the fish’s length”.

22) Line 118-158 – important data – I would like to see a brief description here of how the 3 age groups differed from one another in their behavioural profiles before and after PTZ application –

the description is (understandably) focused on demonstrating a similar response between the 3 but the differences are important for labs using different stages for various reasons

Thank you for the comment. Although we agree that the developmental context is important, a comprehensive developmental comparison is beyond the scope of this study, as we are focused on refining disease models rather than characterizing age specific developments. Our primary goal was to assess whether our new system could sensitively detect seizure severity across different larval stages and this results section captures that. Further, the inclusion of higher-order kinematic measures allowed us to detect and distinguish differences amongst the age groups, particularly in younger, less mature larvae not previously appreciated. However, this manuscript is not intended to guide the selection of developmental time points, but rather to validate a toolset that performs reliably across them.

23) Line 157 – these are all convulsive behaviours are they not – just at different stages – just refer to different stages of convulsive behaviour based on your 2005 paper.

We apologize but simply do not entirely understand the critique as the convulsive stages are already mentioned alongside the behaviors (i.e., whirlpool and convulsive). Further, it is important to decipher the details of these behaviors as per the max speeds, etc. MCAM based measurements of these seizure stages presented here is more complex and detailed than that reported in our original 2005 publication.

24) Line 160 – the kinematic data are key here, and the really novel bit, especially when analysing different types of convulsive behaviours, identifying more subtle impacts of pro and anticonvulsive compounds etc. For me, these need to be incorporated into the ML approach (see below) – they may not be revealing with 15mM PTZ (and perhaps were removed after PCA?), but my guess is that they will become much more important once you start to look at more subtle mechanisms or lower concentrations.

Kinematic data is encoded in and computed from the raw key point values. We use the raw key point values as the input to our machine learning approach so that the model training has access to the rawest form of our measured data. Separately from the machine learning approach we compute kinematics based on our understanding of what is important in the zebrafish behavior. These are two orthogonal approaches but in fact the machine learning implementation has access to much more information than the kinematics which have been abstracted from the raw data. Two sentences have been added to the methods clarifying this point.

25) Line 199 onwards – refer to as Ca²⁺ rather than calcium.

Edits made.

26) Line 201 – Although the Ca²⁺ imaging is elegantly performed, especially with simultaneous LFP recording, I struggle to see how this relates to the abrupt changes in inter-eye distance you describe since you cannot observe this as occurring during a Ca²⁺/ECG spike? Can you make it more clear how this helps.

We respectfully believe the Ca²⁺ imaging data adds an important confirmation, using a novel combination of electrophysiology and non-invasive fiber photometry, that abnormal seizure activity is a CNS-generated phenomenon. The point we were trying to make here is that the qualitative nature of these recording traces over time closely matches those obtained from inter-

eye distance measurement traces. Inter-eye distance in freely swimming zebrafish larvae is a sensitive behavioral proxy for seizure activity, showing strong temporal and functional correlations with calcium transients (via fiber photometry) and forebrain LFP recordings. Abrupt changes in this inter-eye distance measure reflect the motor output of hypersynchronous neural activity, as confirmed by synchronized time-series analysis using calcium dynamics along with neural spike wave discharges. By integrating behavioral, optical, and electrophysiological data, this multimodal approach offers a robust and comprehensive view of seizure dynamics, reinforcing the validity and rigor of our model.

The text (lines 221-224) was edited to clarify this point as follows: “This approach enabled us to capture hypersynchronous neural activity and highlight the similarities with the motor manifestations of this activity exhibited during changes in inter-eye distances”.

27) Also are the ‘twists and turns’ not just a product of animal bouncing off the sides of the well during a convulsion? This also has implications for swim speed – my guess in an open arena swim speeds will be higher than those you state, and this goes back to my thought that larger area would provide more subtle measures of a convulsion.

Seizing larval zebrafish experience rapid twists and turns entirely independent of well size, as these have been observed and reported in a variety of conditions with both small and larger well diameters. The larvae are not simply “bouncing off the sides of the well” during a convulsion as these ‘twist and turn’ behaviors can be also observed in a standard sized Petri dish (*image at right).

*difficult to appreciate in a still image, but more than happy to provide the video file if necessary.

We report speed in a 96-well plate.

28) Often this loss of balance is observed during inter-ictal periods – did you analyse baseline versus treatment period frequency of this as I suspect it happens a lot more during interictal periods than during baseline (or at least the occurrence of this loss of posture as a proportion of time spent not moving)

Post-ictal loss of posture does not occur in larvae that are not having severe seizures.

29) Lines 224-228 – Why were more kinematic endpoints not included in the model in some capacity? I suspect these more subtle and finer scale measures will become more important once you look beyond GABAA antagonism as an initiating mechanism.

By feeding the raw key point data to the model we give it access to all of the information we have and the training process allows the model to make sense of the data as a whole. Kinematic endpoints are computed from the raw key point data based on what we “think” may be important to classify behavior. If we were to input kinematic endpoints into the model this would be identical to us saying that we already have made sense of the data and we know what is important and in doing so we would limit the model to make sense of the data with a new boundary. For example, if we included speed as a feature of our input vector, we would be requiring the model to use this to make it's classifications. By not including speed, we allow the model to decide what features are important itself and maybe it does not need speed. In fact,

because the kinematic endpoints are encoded in the raw key point data, the model does have access to them and much more.

30) Did you assess for dose dependency and if so, what was the threshold for detection against baseline?

Concentration-dependent behaviors seen with varying PTZ doses from 1-20 mM have been studied in many prior publications by our group and others. The “threshold” for Stage 3 convulsions as initially described in our first PTZ publication (PMID: 15730879) is that these seizure events, which are all-or-none phenomena, simply occur less frequently at lower PTZ concentrations. A WT sibling ethogram is now shown as Supplemental Figure 5 for comparison.

31) How about the ability of automated detection of seizures in plate of randomly distributed treated and untreated larvae? What was the detection rate, FPs and FNs?

The detection algorithm was trained on both untreated WT and PTZ-treated WT larvae, with automated seizure detection performed blindly. Randomly selected frames were labeled without knowledge of treatment. The model achieved 87% accuracy (see supplementary figure). For the *scn1lab* dataset, larvae were randomly distributed across wells for both acquisition and analysis, with post-hoc genotyping. The algorithm reliably identifies seizing fish in both examples. Further, for each experiment 96 larvae obtained from Petri dishes containing 100s of larvae, are randomly selected and distributed into 96-well plates for all experiments. This is now noted in the Methods section (line 421).

Discussion

32) Line 267 – did they enable automated detection of complex seizure behaviors in larval zebrafish at scale? I didn’t see application of this, for example by testing in a plate of randomly distributed treated and untreated larvae. If not, this should be rephrased as the possibility to automatically detect – better still test it and include the data – it’s a straightforward and quick experiment.

Our use of the “at scale” terminology refers to the large-scale, simultaneous monitoring of nearly 100 individual larvae, approach that is possible here. This contrasts with similar rodent behavioral tracking studies that can only monitor a single animal (PMID: 36841241). As this approach allows for multiple acquisition experiments, it easily scales up to several 100 larvae per day.

To note, larvae are randomly distributed. Indeed, for genetic epilepsy lines, 96-well plates included randomly distributed WT, heterozygote and homozygote larvae with post hoc genotyping to provide a further level of unbiased analysis.

33) Line 287 – “constrained by smaller arenas or single-subject tracking” Not sure any of these studies used areas smaller than a 96 plate well? – Please clarify - In my opinion a 96 plate well is already too small as discussed above.

Deleted this sentence.

34) 296 – As stated above, I would like to see incorporation of the kinematic endpoints into the ML approach for detection of more subtle events - this needs to be discussed here along with the fact that only one compound/mechanism was tested, no concentration responsiveness was

shown, and also an acknowledgement that the assessment of PTZ-induced convulsions is likely to be at the more severe end of the scale should be made

See comment above about how raw pose estimation data encodes both kinematics and behavior patterns separately.

We did not do concentration responsiveness as 15 mM PTZ is noted in the literature as leading to robust seizure behavior in WT zebrafish, and many studies already tested different PTZ concentrations. More subtle behaviors are noted in the genetic zebrafish models. That this chemoconvulsant is at the more severe end of the scale is now noted as such in the Discussion (line 365).

Methods

35) Line 331 – was the tracking undertaken in the light or dark?

Dark.

36) Line 334 – was thorough mixing achieved?

Yes.

37) The final concentration of PTZ used was high - others have demonstrated behavioural effects at much lower concentrations – why was this concentration selected and did you assess lower concentrations to establish a threshold for detection?

As established in Baraban et al. 2005 (PMID: 15730879) and several hundred subsequent publications, 15 mM PTZ is a standard concentration that consistently elicits larval zebrafish seizures between 3 and 7 dpf. A well-established replication literature on this initial publication exists. As noted above, lower PTZ concentrations simply result in less frequent (but with the same all-or-none durations and convulsive patterns) seizure events.

38) Line 363 – was no chemical used for restraint alongside the agarose?

Correct, only 2% low melting point agarose was used for larval restraint (line 455). No additional chemicals were added.

39) Line 367 – Ca²⁺ rather than calcium

Edited as suggested.

40) Line 363 – what was the contents of the LFP recording chamber?

Embryo media. Edited as follows (line 455): "Recording chambers were bathed in embryo media, placed on the stage of an upright microscope (Olympus BX-51W) and monitored continuously using a Zeiss Axiocam digital camera."

41) Line 405 – ".....considered anomalies and the erroneous frames were excluded from the tracking data". Why were these considered anomalies – was there a statistical threshold you used to objectively discard these data points?

When speeds were plotted as a histogram it was evident that above 120 mm/second the data became quite noisy. We hypothesized that above 120 mm/second was erroneous frames and then verified this by eye looking at frames where this speed was reached.

42) Line 407 – what was the justification for using 0.7x the larval length away from the center of mass (COM) of the larva as the cut off point here?

When all key points are placed correctly on a straight and stationary fish the snout and caudal fin extrema are approximately 0.5x the fish length away from the center of mass. 0.7x adds another 20% on this value for these extreme points.

Figures

43) The figures are very nice and very informative

Thank you.

Reviewer #4 (Remarks to the Author):

In their manuscript titled “Automated detection of complex zebrafish seizure behavior at scale,” Whyte-Fagundes and colleagues employ a multi-camera imaging system enabling high-throughput behavioral assays to track larval zebrafish seizure activity. Additionally, they develop a machine-learning-based algorithm for the offline, automated detection of both normal and abnormal behaviors.

The study is well-designed and potentially valuable for a broad range of researchers using zebrafish as a model organism, beyond epilepsy research alone. However, I have concerns that should be addressed before recommending the manuscript for publication in *Communications Biology*.

Major concerns:

44) The authors should provide a rationale for using 5-minute tracking intervals rather than continuous recording when monitoring motor seizures.

These 5-min continuous recordings were sufficient to monitor and capture spontaneous motor seizures. We used 5-min as that was the current upper limit of the Ramona acquisition software and hardware used to collect such high-resolution data across all 96-wells at 160 fps. A comment on the limitations of computing power was added to the Discussion.

45) How do authors explain the reduction in locomotor activity and total distance traveled between 5 and 7 dpf larvae (Fig. 2a and 2b)?

There is well documented literature to suggest that decreased locomotor activity and total distance traveled in 7 dpf larvae compared to 5 dpf may be due to (i) nutritional transitions and energy availability (yolk sac depletion: PMID 21147203, reduced energy stores: PMID 38787128), (ii) maturation of neural circuits (further development of inhibitory circuits: PMID 17475807, changes in spinal cord circuits related to motor output: PMID 23267317), or (iii) behavioral adaptation (older larvae modulate behavior based upon internal states: PMID 20075256).

46) The manuscript refers to a representative plot for inter-eye distance as Figure 5d (lines 190-191), but panel d is not present in Figure 5. Are the authors instead referring to Figure 6a, or is a figure panel missing?

Thank you for catching this error. It is corrected in text to refer to 6a. No panel was missing.

47) In lines 195–197, the authors claim: “It is not feasible to simultaneously record inter-eye distance and abnormal brain activity in seizing larvae moving at speeds above mm/sec.” This statement is quite strong. Do the authors have evidence to support this? It seems premature to exclude the possibility of achieving this goal using wide-field calcium imaging, light-field microscopy, or tracking microscopy. For context, I encourage the authors to consider the following references:

PMID: 28596596

PMID: 32778840

PMID: 28892088

We read the suggested literature, and the revised manuscript was edited to remove this sentence from the discussion, as the current manuscript does not have sufficient evidence to support this statement.

48) The manuscript notes similarities between baseline/PTZ inter-eye distance traces and those of LFP + photometry (lines 207-209). However, the parallelism is not fully explained. For example, similarities could also exist between brain activity recordings and locomotor activity variables (e.g., distance traveled or heading angle). The authors should clarify this parallelism to ensure it does not come across as a trivial observation (e.g., both traces merely showing episodes of activity interspersed with baseline states).

See response above.

49) The authors claim that machine learning-based detection of motor seizure patterns can serve as a surrogate measure for brain activity (lines 300-302). While every movement undeniably reflects underlying neuronal activation, the inverse is not necessarily true. This point, which links to the previous one, requires clarification.

Revised manuscript was edited to remove this reference to a surrogate measure. Now reads as follows (lines 347-348): “Single-camera systems^{21, 24, 60, 66, 67} utilize simple measurements like total distance moved to capture gross larval seizure behaviors. This approach was confirmed and extended here”.

50) Why was a zebrafish line with nuclear-localized calcium reporters chosen instead of one with cytoplasmic localization, given that fiber photometry captures aggregate, non-cellular resolution measurements? Please provide rationale for this choice.

We employed a GCaMP line commonly used for zebrafish larval imaging studies (PMID: 39746401; PMID: 39367042; PMID: 37689065; PMID: 37019622). As Ca²⁺ fluctuations in the photometry signal were robust and brain-specific, there was no need to use cytoplasmic localization as an additional experiment.

All Minor concerns were addressed.